# Large language models effectively leverage document-level context for literary translation, but critical errors persist

**Marzena Karpinska**     **Mohit Iyyer**
University of Massachusetts Amherst
{mkarpinska, miyyer}@cs.umass.edu
https://litmt.org/

## Abstract

Large language models (LLMs) are competitive with the state of the art on a wide range of *sentence-level* translation datasets. However, their ability to translate paragraphs and documents remains unexplored because evaluation in these settings is costly and difficult. We show through a rigorous human evaluation that asking the GPT-3.5 (text-davinci-003) LLM to translate an entire *literary* paragraph (e.g., from a novel) at once results in higher-quality translations than standard sentence-by-sentence translation across 18 linguistically-diverse language pairs (e.g., translating into and out of Japanese, Polish, and English). Our evaluation, which took approximately 350 hours of effort for annotation and analysis, is conducted by hiring translators fluent in both the source and target language and asking them to provide both span-level error annotations as well as preference judgments of which system's translations are better. We observe that discourse-level LLM translators commit fewer mistranslations, grammar errors, and stylistic inconsistencies than sentence-level approaches. With that said, critical errors still abound, including occasional content omissions, and a human translator's intervention remains necessary to ensure that the author's voice remains intact. We publicly release our dataset and error annotations to spur future research on the evaluation of document-level literary translation.[1]

## 1 Introduction

Large language models (LLMs) such as ChatGPT (OpenAI, 2022) demonstrate remarkable performance as stand-alone translation systems, rivaling and sometimes surpassing commercial models on sentence-level benchmarks (Vilar et al., 2022; Hendy et al., 2023; Jiao et al., 2023). Furthermore, LLMs are increasingly being deployed for *document-level* translation (Book Maker, 2023;

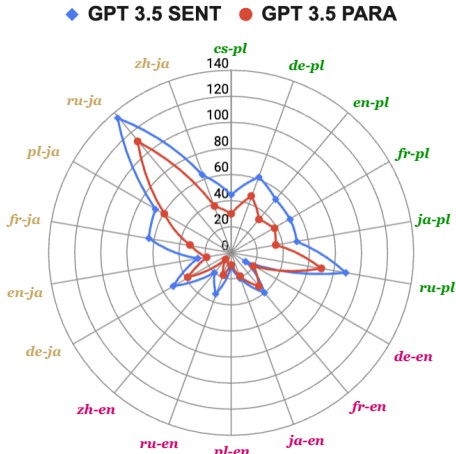

Figure 1: A plot of the total number of errors annotated in sentence-level (SENT) and paragraph-level (PARA) translations produced by GPT-3.5 across 18 different language pairs. In all cases, PARA produces fewer errors than SENT, which demonstrates that GPT-3.5 takes advantage of discourse context during translation.

Pawlak, 2023), a scenario for which there are currently no reliable automatic evaluation methods. In this paper, we hire human translators to conduct a rigorous fine-grained evaluation of GPT-3.5's ability to translate **paragraph-level** texts from **literary works** across 18 different language pairs. Our results (Figure 1) demonstrate that GPT-3.5[2] effectively leverages discourse-level context to produce higher-quality translations than when translating sentences in isolation.

**Why literary texts?**   Translating works of literature poses unique challenges due to the intricate nature of creative work and the importance of capturing the author's voice and contextual nuances. Translators thus apply a wide range of transla-

---

[1] https://github.com/marzenakrp/LiteraryTranslation

[2] We completed our annotations on translations from the text-davinci-003 checkpoint obtained prior to the API release of ChatGPT and GPT-4. Nevertheless, we include a preliminary analysis of GPT-4's translations in §F.

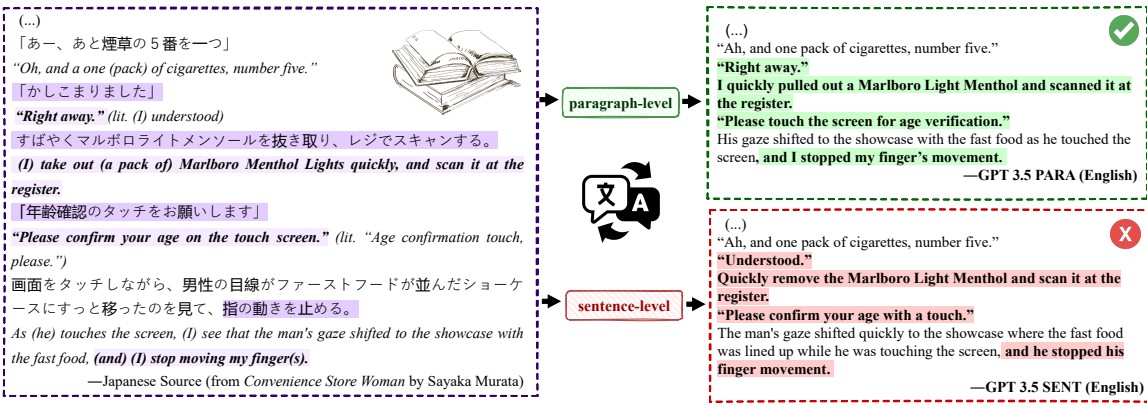

Figure 2: An example of paragraph-level (PARA) and sentence-level (SENT) translations of the same Japanese paragraph into English. Sentence-level translation results in a range of erroneous translations, from worse word choice ("understood" vs "right away") to incorrect pronouns ("he" vs "I"); these errors are corrected by PARA.

tion techniques (Chesterman, 1997; Molina and Hurtado Albir, 2004), from simple shifts in grammatical categories to more complex stylistic or content-based rearrangements that often cross sentence boundaries. Translators may also merge or split sentences and paragraphs, which renders the traditional sentence-level pipeline insufficient for capturing the full scope of the original text (Toral and Way, 2015; Taivalkoski-Shilov, 2019b; Post and Junczys-Dowmunt, 2023; Jiang et al., 2023).[3] Taken together, these properties make literary texts a good testbed for document-level machine translation (Thai et al., 2022); in our work, we focus on the *paragraph*[4] as a minimal discourse-level unit.

**Why human evaluation?** The absence of rigorous document-level evaluations of LLM translators is striking but also somewhat understandable given the unreliability of automatic metrics (Thai et al., 2022) and the difficulty of properly conducting human evaluations (Castilho, 2021). Furthermore, evaluations of LLM translators are especially difficult due to data contamination (Aiyappa et al., 2023; Chang et al., 2023), as it is unclear whether the models are pretrained on existing benchmarks (e.g., from WMT). We fill this gap by first collecting paragraphs from recently-published literary translations. Then, we provide human translators with two candidate machine translations of a given source paragraph and ask them to (1) mark error *spans* and categorize them based on a predefined

schema inspired by MQM (Lommel et al., 2014b; Freitag et al., 2021), (2) make preference judgments of which of the two translations is of higher quality, and (3) provide free-form justifications of their preference judgments. In total, we collect such annotations on **720** pairs of translated paragraphs across **18** different language pairs (using three diverse target languages of English, Japanese, and Polish), which we then leverage for a fine-grained analysis of the behavior of different LLM translation methods.

**LLMs produce better translations when provided with paragraph-level context:** Our evaluations reveal that using GPT-3.5 to translate complete paragraphs via few-shot prompting (PARA) yields translations of significantly higher quality than both the sentence-by-sentence GPT-3.5 methods (SENT, PARA_SENT) as well as Google Translate. Our detailed analysis of annotated translation errors and free-form comments shows that PARA exhibit increased coherence, better preservation of literary style, and improved handling of context-dependent expressions (see Figure 2). That said, PARA makes many critical mistranslations and other errors across different language pairs, which shows that LLM-based translators still have significant room to improve, particularly when translating contextually-rich literary texts.

## 2 Background

Before describing our dataset and evaluation, we first contextualize our work within the recent body of research on translation via large language models. We also survey the broader body of document-

---

[3] At least 55% of the reference target paragraphs used in our study split or merge sentences from the source text (measured with an automatic sentence tokenizer).

[4] We broadly define a paragraph as a distinct passage within the novel, focusing on a single theme.

level[5] MT research in §A.

**Translation with large language models:** LLM-based translation is attractive because a single model, without training or fine-tuning on *large* parallel corpora, can produce high-quality translations across many language pairs.[6] Recent work explores LLMs' capabilities in this space (Wang et al., 2023) spanning paragraph-level post-editing with LLMs (Thai et al., 2022), translating sentence-level inputs (Vilar et al., 2022; Jiao et al., 2023), analyzing hallucinations in LLM-generated translations (Guerreiro et al., 2023), and employing LLMs to evaluate machine translation (Kocmi and Federmann, 2023). Simple sentence-level English prompt templates have been found effective for paragraph translations (Zhang et al., 2023), and automatically-generated dictionaries can assist LLM-based translation (Ghazvininejad et al., 2023; Lu et al., 2023) along with selecting high-quality demonstrations (Vilar et al., 2022). To the best of our knowledge, the only prior work other than ours that evaluates LLMs for paragraph-level translation is Hendy et al. (2023), who conduct *automatic* evaluation of context-aware sentence-by-sentence translation; in contrast, we perform a fine-grained *human* evaluation of paragraph-level translation.

## 3 Data & methods

Our work differs from existing research on translating with large language models in two key ways: we focus on translating *literary* text at the *paragraph level*. In this section, we describe and motivate the paragraph-level translation dataset used in our study, which covers 18 unique language pairs (three target languages) and is sourced from recently-published novels. Then, we outline the different ways in which we leverage GPT-3.5 to translate these paragraphs at both the sentence and paragraph levels.

### 3.1 Dataset collection

Literary texts (e.g., novels or short stories) pose unique challenges for translators due to their complex nature. Translators must interpret and honor the author's voice with no objective reality to measure against, which can result in several equally

valid translations (Sager, 1998). For machine translation systems, these challenges exacerbate the need for discourse-level context (Thai et al., 2022): an author's intended meaning or style is often unclear from just a single sentence.

**Selecting paragraphs from novels:** How good are machines at translating literary paragraphs? To answer this question, we extract **20** paragraphs (dialogues and narrative texts) each from **18** recently-published translations of novels, and we manually align these paragraphs with corresponding paragraphs in the source novel[7] (see Table 8 in §B). Almost all of the translations were published after 2021 (see Table 7 in §B), which is important to avoid data contamination with LLM pretraining data (Aiyappa et al., 2023; Chang et al., 2023). In sum, we obtain **360** aligned source-target paragraphs, which we use for all of the experiments described in the rest of the paper.

**Data memorization issue:** In order to investigate the extent to which `text-davinci-003` may have memorized the novels in our dataset, we employ the prompts from (Chang et al., 2023) and assess the model's ability to produce masked characters' names. For this purpose we select 171 translation paragraphs, which contained character's names, resulting in an average of 8 out of 20 paragraphs used per book. In nearly all instances, the model was unable to accurately produce the correct names, with three exceptions. Two of these were names of well-known historical figures, "Napoleon Bonaparte" and "Simonides of Ceos." A closer examination revealed that these names could likely be inferred from the context, rather than being a result of the model's memorization. In the third instance the model produced the correct name but in diminutive instead of augmentative form ("Kasia" instead of "Kaśka").[8]

Additionally, we tested `text-davinci-003` with a randomly selected subset of paragraphs from our dataset. In these cases, the model was unable to generate accurate completions.

---

[5] Note that the term "document-level" has been used in MT research to denote both multi-sentence passages as well as complete documents.

[6] That said, parallel data is almost certainly included in LLM pretraining data, at least for high-resource languages (Briakou et al., 2023).

[7] We purchase the source ebook and its corresponding translation before extracting aligned paragraphs.

[8] Kasia/Kaśka" are both forms of "Katarzyna," the second most common female name in Poland as of January 2023. This raises a question of whether the model's response was due to memorization or an educated guess based on the name's popularity (https://www.statista.com/statistics/1089014/poland-most-popular-female-names/).

**Paragraph length:** All paragraphs consist of at least two sentences, and the majority of them are between four to nine sentences long (mean=7.45, std=4.14).[9] As automatic sentence tokenizers are not always reliable for all of the languages considered in our study, we manually perform sentence tokenization to enable a direct comparison of sentence and paragraph-level translation systems. For more details about the dataset statistics, including token and sentence counts, see §B, which also includes data on sentence numbers obtained using a sentence tokenizer.

**Source and target languages:** As source languages, we select eight languages that belong to different language families, have varied morphological traits, and employ different writing systems: English (*en*), Polish (*pl*), Russian (*ru*), Czech (*cs*), French (*fr*), German (*de*), Japanese (*ja*), and Chinese (*zh*). As target languages, we select English, Japanese, and Polish, as they also vary greatly in their morphology, grammar, and writing systems. The detailed rationale can be found in §B.

### 3.2 Translation with large language models

In this paper, we focus on translating the literary paragraphs in our dataset using large language models. More specifically, we use the GPT-3.5 `text-davinci-003` checkpoint, which has been further tuned to follow instructions based on human feedback (Ouyang et al., 2022). Hendy et al. (2023) demonstrate that GPT-3.5 produces translations of reasonable quality, though their focus was mainly at the sentence level. Since many LLMs, including GPT-3.5, are only accessible via black-box APIs, we adapt the model for translation via in-context learning (Brown et al., 2020).

**Demonstration examples:** We use few-shot prompting, in which a model is provided with a prompt consisting of five demonstrations. We manually curate the five demonstrations from literary texts for each of the 18 language pairs, resulting in 90 total demonstration examples. These demonstrations are sourced from novels that are *not* part of our translation dataset, resulting in potential differences in topic and style (see Table 9 in the §B for details). We further ensure that each set of

five demonstrations includes both dialogues and narrative texts.

**Prompting for translation:** We consider the following three prompting strategies for GPT-3.5 that allow us to compare the model's abilities to translate with and without discourse-level context (see Table 1 for templates and §C for the exact prompts):

- **GPT-3.5 sentence-level translation without context (SENT):** Each sentence of the paragraph is translated in isolation of the others. To maintain consistency, we provide the same five *sentence*-level examples[10] in each prompt for the given source-target language pair.[11]

- **GPT-3.5 sentence-level translation *with* context (PARA_SENT):** Each sentence of the paragraph is translated in context. The model is provided with the entire source paragraph as input, where the sentence to be translated is wrapped in `<translate>` and `</translate>` tags, in addition to a partially-translated target paragraph. The demonstrations are also presented with the same tags. For each demonstration in the prompt, a sentence in a different position was chosen (e.g., from the beginning, middle, and end of the paragraph).

- **GPT-3.5 paragraph-level translation (PARA):** The entire source paragraph is passed into the model, and the output target paragraph is generated conditioned on this input (i.e., without any sentence tokenization). Demonstrations in the prompt are also *paragraphs*[12] of translations from the respective source language into the target language in question.[13]

---

[9]A paragraph with fewer sentences is not necessarily short: for example, in the German novel "An Inventory of Losses," sentences can be as long as 70 to 80 words, with the longest reaching 117 words. The distribution of sentences in paragraphs is provided in Figure 7 in §B.

[10]Sentence-level demonstrations for SENT are sampled from the demonstrations for paragraph-level translation.

[11]To ensure consistent quotation mark usage and enable a fair comparison with paragraph-level translations, quotation marks in sentence-level translations were manually adjusted.

[12]The examples for PARA and PARA_SENT configurations are necessarily lengthier. Due to the GPT-3.5 maximum context size, it is not always possible to include all five examples within the prompt. Consequently, around 10% of the data was translated using four or fewer examples.

[13]Initially, we experimented with GPT-3 by translating between two non-English languages using English as a pivot, as it is the primary language of the model. The model had access to the source text and its English translation. After manual evaluation and comparison to translations without a pivot language, we found no significant benefit in using English as the pivot. Consequently, we directly translated paragraphs into the target language. Refer to §H for details and results of this preliminary study.

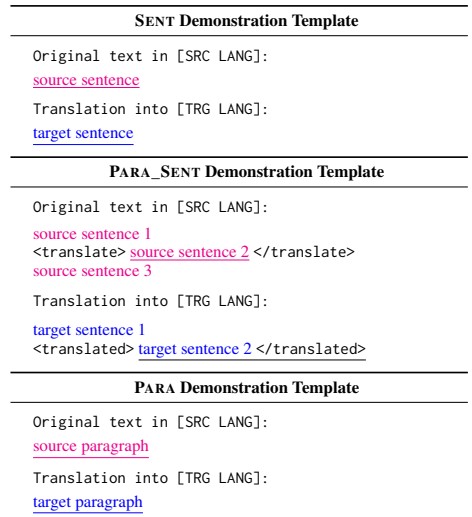

Table 1: Prompt templates for Sᴇɴᴛ, Pᴀʀᴀ_Sᴇɴᴛ, and Pᴀʀᴀ. The source text to translate and expected target outputs are underlined.

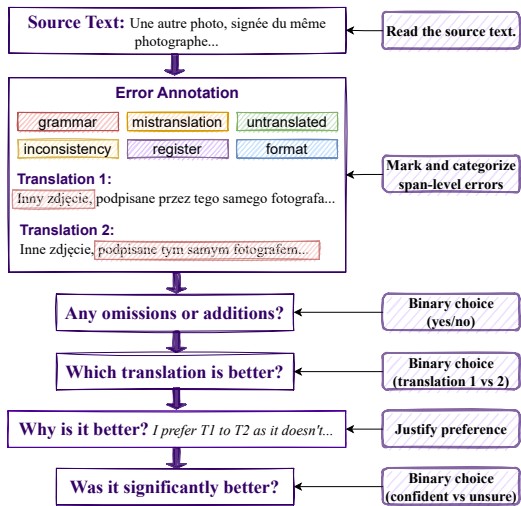

Figure 3: A description of the annotation process for a pair of candidate translations given a source paragraph. Note that our hired translators go through this pipeline for *three* different pairs per source paragraph, comparing Pᴀʀᴀ with Sᴇɴᴛ, Pᴀʀᴀ_Sᴇɴᴛ, and GTʀ.

**Using Google Translate (GTʀ) as a baseline:** In order to compare commercial-grade translation systems to LLM translators, we also translate all paragraphs in our dataset using Google Translate.[14] We opt for an off-the-shelf commercial system instead of a state-of-the-art system from, for instance, WMT competitions for two primary reasons. First, our experiments focus on *literary* translations. Given that WMT systems are predominantly evaluated on the news domain, it is uncertain which system would perform best, and some language pairs may not even be supported. Second, our main research question revolves around LLMs' ability to incorporate contextual information, rather than merely comparing their performance with state-of-the-art translation systems. We employ GTʀ as a reasonably robust baseline to assess the extent to which context can enhance MT quality, rather than asserting that LLMs outperform *all* traditional MT systems.

## 4 Evaluating document-level literary translation

How do we compare the translation quality of the systems described above? Automatic metrics such as Bʟᴇᴜʀᴛ and Cᴏᴍᴇᴛ are untested on document-level inputs as well as literary texts, and as such we do not consider them reliable, although we do

report them in §G.[15] Human evaluation is equally problematic, as direct assessments of translation quality (e.g., "rate the quality of this translation from 0-100") suffer from calibration issues that are exacerbated with longer texts (Karpinska et al., 2021). Thus, we opt for a human evaluation inspired by Multidimensional Quality Metrics (Lommel et al., 2014b, MQM), in which annotators mark and classify error spans within the translation. Specifically, for each of the 18 language pairs studied in this work, we hire translators to identify all span-level errors in two competing translations. For each evaluated pair, the annotators were also asked to choose the better translation and provide a free-form rationale. For each source paragraph, the translators make three binary judgments of which translation is higher quality: Sᴇɴᴛ vs Pᴀʀᴀ, Pᴀʀᴀ_Sᴇɴᴛ vs Pᴀʀᴀ, and GTʀ vs Pᴀʀᴀ.

**Recruiting annotators:** As our task is complex and requires fluency in both the source and target language, we hire *translators* to provide the annotations. We recruit 13 translators via the Upwork freelancing platform,[16] each of whom is a native speaker of English, Polish, or Japanese.[17] One

---

[14]All paragraphs were translated in January 2023 using the GoogleTranslate API. The system was provided entire paragraphs, which it likely partitioned and translated sentence-by-sentence.

[15]Automatic metrics developed specifically for document-level MT are also insufficient as they either work best with one-to-one sentence level alignments (Vernikos et al., 2022; Hendy et al., 2023) or are available only for English (Jiang et al., 2022).

[16]https://www.upwork.com/

[17]The annotators for Czech-Polish and Russian-English were both native speakers of the respective source languages

translator, hired directly, was a bilingual speaker of English and Polish with advanced knowledge of German; as such, she performed the *pl-en*, *de-en*, and *de-pl* evaluations. Evaluation of *ja-pl*, *pl-ja*, and *pl-en* texts was done by the first author in a collaboration with native speakers of Polish/Japanese to avoid any potential bias. Each translator was paid $2 per evaluated pair of candidate translations, with an additional $5 bonus to cover the time spent familiarizing themselves with the instructions. We asked them to compare three pairs of system translations (PARA vs. SENT, PARA vs. PARA_SENT, PARA vs. GTR) for 10 paragraphs per language pair[18]; as such, 180 total source paragraphs were used in our evaluations. Altogether, we paid approximately $12 per hour, with a total cost of $955.

**Annotation task:** First, we tasked the hired translators[19] with annotating a subset of MQM translation errors identified through a pilot analysis and annotation of the system's outputs. Specifically, we ask them to highlight spans within the candidate translations that contain errors belonging to any of the following error categories:

- **mistranslation:** [20] accuracy errors that occur when the wrong target word or phrase is chosen to represent content from the source text. In addition to canonical mistranslations, we also include *overly literal* translation errors that occur when systems nonsensically translate word-by-word into the target language.

- **grammar:** grammatical errors, such as errors in conjugation, declension, or wrong prepositions.

- **untranslated:** words or phrases that should have been translated into the target language but were either left in the source language or just transliterated into the target language.

- **inconsistency:** use of different terms to refer to the same entity, or different words where the same word should be used for stylistic reasons (e.g., "Kasia" and "Kate," "coat" and "jacket," or "bad" and "awful" ).

- **register:** a clear violation in the use of formal and informal language within the same text, only annotated in Japanese.[21]

- **format:** incorrect usage of punctuation (e.g., "." instead of "。").

After the span-level annotation is complete, we then ask the translators to further identify if any of the candidate translations contains significant content **additions** or **omissions** in relation to the source text.[22] Finally, they are asked to **choose the better translation** and provide a justification for their choice in two to five sentences. We instruct them to additionally mark whether their chosen translation is significantly superior, or if the decision was difficult because both translations are of roughly comparable quality (see Figure 3 and §D for details).

## 5 Results

In this section, we compare our different literary translation methodologies using both automatic metrics and aggregate statistics from the human evaluations. Overall, we observe that the PARA configuration outperforms competing methods across all evaluations and language pairs. These results demonstrate that GPT-3.5 effectively leverages paragraph-level context to produce better translations than sentence-level methods, and also that the less efficient sentence-by-sentence translation with context is (PARA_SENT) is unnecessary to achieve high translation quality.

### 5.1 Human evaluation also favors PARA

Figure 5 contains human preference results comparing PARA to SENT, PARA to PARA_SENT, and

---

[18]These paragraphs were randomly sampled from the 360 paragraphs. The entire set of 360 paragraphs was used for the automatic evaluation described in §G.

[19]They were presented with guidelines in their native language. The annotation task was performed using the Label-Studio annotation tool (Tkachenko et al., 2020-2022). See Figure 11 for the screenshot of the interface.

[20]We note that mistranslations in literary text are often not as grave as, for instance, in news articles. Human translators hold *poetic license*, which allows them to change some details to make the text more enjoyable for the reader. Is changing "bonito" into "tuna" incorrect? Or can it be perceived as a way to accommodate an English-speaking readership that is likely more familiar with the latter?

and highly proficient in their respective target languages. They collaborated with native speakers of the target languages, who possessed a basic understanding of the source language, to complete their annotations.

[21]We only annotate cases where the level of formality changes abruptly within the same paragraph. It is possible that a given character would be more likely to use formal language but an informal language is being employed. As long as this is consistent we do not consider it an error as this cannot be fully determined from the paragraph context.

[22]Note that this task was simplified to a binary choice – either there were serious omissions/additions or not. We did not ask the annotators to further annotate them due to the time restrictions.

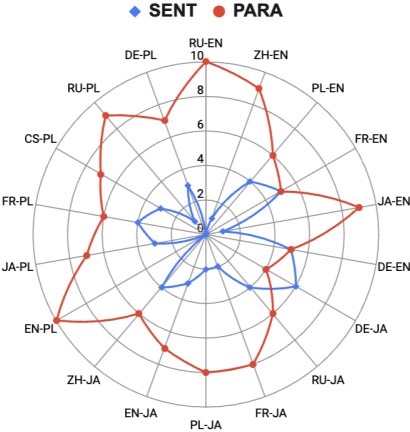

Figure 4: The distribution of translator preference judgments between sentence-level translation (SENT) and paragraph-level translation (PARA). PARA is preferred (i.e., more votes) in every language pair except *de-ja*, *fr-en* and *de-en*.

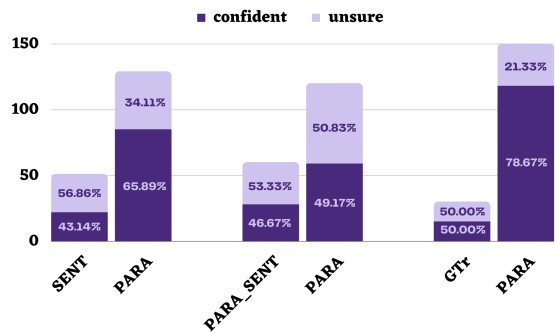

Figure 5: The number of votes for SENT vs PARA, PARA_SENT vs PARA, and GTR vs PARA along with rater confidence (*confident* or *unsure*). PARA is preferred to all competing methods. All differences are statistically significant at *p*<.001 (binomial test).

PARA to GTR, aggregated across all 18 language pairs studied in this paper (i.e., 180 votes per system comparison). Table 11 breaks down these results for each language pair, and we observe the same trends for the vast majority of pairs. Overall, the translators significantly favored PARA translations over the alternatives (*p*<.001, binomial test). Table 2 contains specific information about grammar and mistranslation errors split across the three target languages (see Table 16 and Table 17 for details), which we refer to in the discussion below.

**PARA is clearly better than SENT:** PARA is preferred by translators over SENT at a rate of 71.67% (*p*<.001, 95% CI [0.645, 0.781]). Additionally, when translators preferred PARA, they were usually confident in the decision (i.e., it was clearly better than SENT); even if we exclude all "unsure" votes, the preference for PARA translations remains significant at 79.44% (*p*<.001, 95% CI [0.705, 0.866]). The only language pair in which SENT is favored over PARA is *de-ja* (see Figure 4).[23] Overall, SENT produces 31% more mistranslations, 48.6% more grammar errors, 15 times more inconsistencies, and 3.5 times more register errors (Table 2).

**PARA is clearly better than GTR:** PARA translations are overwhelmingly preferred over those from Google Translate (GTR), with an 83.33% preference rate (*p*<.001, 95% CI [0.771, 0.885]). In the

fr-ja, pl-ja, zh-ja, and cs-pl language pairs, PARA received *all* of the ten votes over GTR. Overall, GTR translations result in 58.18% more mistranslations, 35.24% more grammatical errors, over seven as many inconsistency errors, and ten times more register errors (see Table 2). §E contains more fine-grained comparisons of these two systems.

**PARA is slightly preferred over PARA_SENT:** Our evaluations show that PARA is better than PARA_SENT, but the gap is smaller than it is for the other two methods. PARA is still preferred at a 66.67% rate (*p*<.001, 95% CI [0.593, 0.735]). Both PARA and PARA_SENT produce a comparable number of mistranslations (*483* vs 462), grammar errors (*105* vs *113*), and inconsistencies (*2* vs *3*) (see Table 2). While PARA_SENT leaves around 22% more words untranslated, it appears to leverage the contexts and even occasionally selects better equivalents in the target language, as evidenced by translator comments. One major issue with PARA_SENT is that it occasionally repeats sentences, whereas PARA never does so.

## 6 Analyzing translation errors

The aggregate statistics from the previous section confirm that PARA-level translation via GPT-3.5 is the strongest literary translator of the methods that we study. Translations produced by PARA are favored by both automatic metrics and human translators, and it makes fewer errors than competing methods. In this section, we dive deeper into specific *types* of errors that are made within each high-level category (e.g., grammar, mistranslation), and we present examples of errors associated with

---

[23]This could be because the German novel *An Inventory of Losses* in our dataset contains the longest sentences of any book (45 tokens per sentence), and thus the intra-sentence context is likely more informative than in other books.

| Type | Trg Lang | Para | Sent | Para_Sent | GTr |
|------|----------|------|------|-----------|-----|
| Mistranslation | En | 88 | 109 | 82 | 155 |
| | Ja | 224 | 295 | 223 | 334 |
| | Pl | 171 | 229 | 157 | 275 |
| | Total | *483* | *633* | *462* | *764* |
| Grammar | En | 5 | 20 | 9 | 18 |
| | Ja | 43 | 49 | 38 | 65 |
| | Pl | 57 | 87 | 66 | 59 |
| | Total | *105* | *156* | *113* | *142* |
| Inconsistency | En | 0 | 5 | 0 | 1 |
| | Ja | 1 | 7 | 2 | 7 |
| | Pl | 1 | 19 | 1 | 7 |
| | Total | *2* | *31* | *3* | *15* |
| Untranslated | En | 13 | 5 | 14 | 6 |
| | Ja | 23 | 30 | 33 | 24 |
| | Pl | 23 | 16 | 25 | 4 |
| | Total | *59* | *51* | *72* | *34* |
| Register | En | 0 | 0 | 0 | 0 |
| | Ja | 7 | 25 | 13 | 71 |
| | Pl | 0 | 0 | 0 | 0 |
| | Total | *7* | *25* | *13* | *71* |
| Format | En | 0 | n/a | n/a | 1 |
| | Ja | 0 | n/a | n/a | 117 |
| | Pl | 0 | n/a | n/a | 8 |
| | Total | *0* | *n/a* | *n/a* | *126* |

Table 2: Total counts of all of the types of mistakes made by each of the four systems from our annotation. Overall, models with access to paragraph-level context commit fewer translation errors.

lack of context understanding made by SENT and GTR that are fixed by PARA.

## 6.1 Language-specific grammatical errors

We analyze the types of grammatical errors that are made by the studied translation methods in all three target languages.[24] In summary, although GPT-3.5 is primarily trained on English, it is competitive with GTR at Polish and Japanese grammar proficiency. In fact, PARA generates the fewest grammatical errors of any system, with a total of *97* for both languages, in contrast to *136* errors made by SENT, *101* errors by PARA_SENT, and *122* errors by GTR (see Table 2). That said, *none* of these systems delivers translations devoid of grammatical inaccuracies, even for English.

**English:** Perhaps not surprisingly, translations into English contain fewer grammatical mistakes than Japanese or Polish (see Table 2). The most prominent mistakes in English are incorrect articles, which is most frequent with SENT and GTR. This is to be expected, as the choice between the definite and indefinite article in English depends heavily on the context. Other mistakes include wrong or omitted prepositions, wrong parts of speech, and incorrect word order (see Table 17).

---

[24]There are some differences in the paragraph lengths between the three target languages that should be taken into consideration when analyzing raw numbers. However, the general tendencies remain intact.

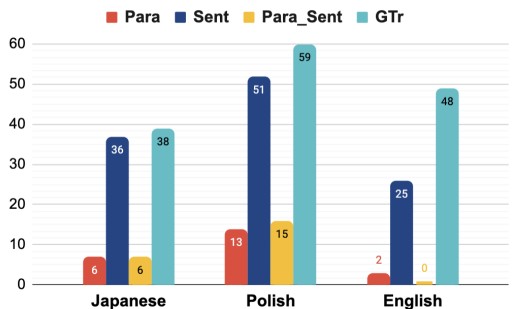

Figure 6: Quantification of mistranslations resulting from missing or misinterpreted paragraph-level context in PARA, SENT, PARA_SENT, and GTR systems, organized by the target language (Japanese, Polish, and English).

**Japanese:** Translations into Japanese contain considerably more mistakes. Most notably, the systems struggle with the correct choice of particle: PARA and SENT produce twice as many mistakes in this regard than PARA_SENT and GTR (see Table 17). Other mistakes include incorrect tense, verb finite form within the sentence, or incorrect word order, the latter of which is much more frequent in GTR than any of the GPT-3.5 translations.

**Polish:** GPT-3.5 exhibits more difficulty with Polish grammar than English or Japanese across all prompting strategies (see Table 2). It frequently generates incorrect gender, case, or prepositions (see Table 17). We also observe instances in which GPT-3.5 alters the gender of a noun, such as producing *grilla*, a non-existent feminine form, in place of the masculine *grill*, while accurately modifying all adjectives and verbs to match the novel feminine noun.[25]

## 6.2 Context-related errors

We manually classify *all* annotated mistranslations (2,324 instances) into subcategories, several of which include instances where the absence of discourse-level context is clearly a contributing factor (see Table 16 for detailed classification).[26] We also further analyze *all* translations in terms of content-related issues. Overall, we observe that context is indeed incorporated into the translations

---

[25]It is worth noting that *grilla* can also be also the genitive form of the masculine noun *grill*; however, the agreement of surrounding verbs and adjectives with the feminine noun suggests that the system likely treated the word as feminine.

[26]The initial classification was conducted on the first version of the dataset. After incorporating small corrections, we identified 18 more mistranslations that were not part of this analysis.

| Type | Source | GPT-3.5 Sent Translation | GPT-3.5 Para Translation | Comment |
|---|---|---|---|---|
| Pronouns | Романы, как известно, печатались на разной бумаге [*paper*]. И гореть она [*she*] может по-разному.

—Russian Source (from *Manaraga*) | Romany, jak wiadomo, drukowano na różnym papierze [*paper*]. I może ona [*she*] tęsknić na różne sposoby.

—GPT-3.5 Sent (Polish) | Jak wiadomo, powieści drukowano na różnym papierze [*paper*]. I może on [*he*] palić się na różne sposoby.

—GPT-3.5 Para (Polish) | "Paper" is a feminine noun in Russian and referred to as "she," whereas it is a masculine noun in Polish and should be referred to as "he," as in Para. The absence of context in Sent leads to an incorrect translation. |
| Cultural Nuances | 「気が付かなくてすみません」「いやいや、(...)。古倉さんは毎日勤務 なのに手を抜かないからねー！」
[lit. *Ms. Furukura works every day*]

—Japanese Source (from *Convenience Store Woman*) | "I'm sorry I didn't notice." "No, no, (...). Furukura-san works hard every day without taking any shortcuts!"

—GPT-3.5 Sent (English) | "I'm sorry I didn't notice." "No, no, (...). You work every day, but you never slack off!"

—GPT-3.5 Para (English) | "Furukura-san" or "Miss Furukura" in the last source sentence is used instead of the second-person "you" as per Japanese convention. Translating this sentence without context into English results in a confusing translation (Sent) that implies that the speaker refers to some other "Furukura" rather than their listener. Para correctly translates "Furukura" as "you." |
| Ellipsis | „Ne, teď uděláš nádobí!" [*(you) will do the dishes!*] „Neudělám!" [*(I) won't do!*] „Uděláš!" [*(You) will do!*]

—Czech Source (from *Crows*) | — Nie, teraz zrobisz zmywanie! [*(you) will do the washing*] — Nie zrobię! [*(I) won't do!*] — Zrobisz to! [*(You) will do it!*]

—GPT-3.5 Sent (Polish) | — Nie, teraz umyjesz naczynia! [*(You) will wash the dishes*]! — Nie umyje [*(I) won't wash*]! — Umyjesz! [*(You) will wash*]!

—GPT-3.5 Para (Polish) | Czech uses the same collocation as English, "*do* the dishes," which is invalid in Polish. Hence, the ellipses in the last two sentences in the source text require a broader context to be translated correctly. Para does it properly, translating both as "wash," while Sent unsurprisingly fails to choose the correct collocation. |
| Subject Ellipsis | When we were done, the lipstick went back into some mother's Fendi handbag. We watched her apply it, unaware.

—English Source (from *A Children's Bible*) | Gdy skończyliśmy, szminka wróciła do jakiejś torebki Fendi należącej do matki. Patrzyliśmy, jak to robi, nieświadomi [*unaware (we)*] tego.

—GPT-3.5 Sent (Polish) | Kiedy skończyliśmy, szminka wróciła do jakiejś torebki Fendi matki. Patrzyliśmy, jak ją nakłada, nieświadoma [*unaware (she)*] naszych działań.

—GPT-3.5 Para (Polish) | Only from the broader context we can deduce that "unaware" refers to the mother, not the "we" (referring to children) watching her. Para correctly attributes the state of being "unaware" to the mother, which is exhibited by its usage of the singular feminine form of the adjective. In contrast, Sent mistranslates it using the plural masculine form of the adjective "unaware," which implies that it refers to "we" rather than the "mother." |
| Consistency | Alles zu vergessen, ist gewiss schlimm [*bad*]. Noch schlimmer [*worse*] ist, nichts zu vergessen (...).

—German Source (from *An Inventory of Losses*) | すべてを忘れることは確かに悲惨な[*tragic*]ことです。さらに悪い[*worse*]のは、何も忘れないことです。

—GPT-3.5 Sent (Japanese) | すべてを忘れることは確か に悪い[*bad*]ことです。もっと悪い[*worse*]ことは、何も忘れないことです。

—GPT-3.5 Para (Japanese) | The German source translates into English as "To forget everything is *bad*, certainly. *Worse* still is to forget nothing." It is arguably important for the translation to repeat the same word which is an equivalent of the German "schlimm" ("bad"). Para does it well, translating both as 悪い, or "bad," while Sent uses two different words, "tragic" and "bad" which results in inconsistent translation. |
| Polysemy | Все прошло хорошо. Книга прочитана идеально – не быстро и не медленно, минимум дыма. Классика. Я был в форме [*in shape*].

—Russian Source (from *Maranaga*) | Wszystko poszło dobrze. Książka została przeczytana idealnie – nie szybko i nie wolno, minimalna ilość dymu. Klasyka. Byłem w mundurze [*in uniform*].

—GPT-3.5 Sent (Polish) | Wszystko poszło dobrze. Książka przeczytana idealnie – nie szybko i nie wolno, minimalna ilość dymu. Klasyka. Byłem w formie [*in shape*].

—GPT-3.5 Para (Polish) | The ambiguity stems here from multiple meanings of the Russian noun форма , which can mean either "shape" or "uniform." Since one can be "in shape" as well as "in a uniform", only from the context it becomes clear which meaning was intended by the author. Para translates it correctly as "shape" while Sent mistranslates it as "uniform." |
| Appropriateness | 「あー、あと煙草の5番を一つ」「かしこまりました」 [lit. *(I) understood*]

—Japanese Source (from *Convenience Store Woman*) | "Ah, and one pack of cigarettes, number five." "Understood."

—GPT-3.5 Sent (English) | "Ah, and one pack of cigarettes, number five." "Right away."

—GPT-3.5 Para (English) | This conversation is between a clerk and a customer. The Japanese expression かしこまりました is an honorific that literally means "understood." However, when choosing the best equivalent, the translator needs to consider the situation at hand to best reflect its meaning in the target language. "Understood" in Sent is technically correct, but it is an unfortunate word choice for the clerk to employ. On the other hand, "right away" in Para fits much better in the context of this conversation. |

Table 3: Examples of different context-related issues observed in Sent translations, which are fixed in the corresponding Para translations. Phrases that exemplify these issues are highlighted in purple, and English glosses are provided in [*square brackets*].

for both Para and Para_Sent outputs, which results in fewer context-dependent issues (see Figure 6). More specifically, we observe that Para produces translations that leverage the context resulting in mostly correct translations of pronouns, ellipsis, cultural nuances, and polysemous words and phrases; Table 3 contains specific examples and discussion of each. Para is also more consistent and appropriate in vocabulary usage than Sent. All cases are further analyzed in §E.2.

## 7 Conclusion

In this paper, we demonstrate that LLMs leverage paragraph-level context to produce translations that are more coherent and enjoyable than sentence-by-sentence translation while containing fewer mistranslations and grammatical issues. Our evalu-

ations reveal that professional translators prefer paragraph-level translations over both sentence-level translations produced by the same language model, and also to those generated by an off-the-shelf commercial system (GTR). We release our dataset and error annotations to help facilitate the development of new evaluation methodologies and automatic metrics for document-level machine translation. Finally, a full-length novel extends far beyond the confines of paragraph-level translation. In future work, we will focus on integrating individual paragraphs into cohesive chapters, which can then be expanded to encompass the entire novel.

## 8 Limitations

So far, we have shown that GPT-3.5 leverages paragraph-level context to produce translations that

are better than those produced by sentence-level counterparts (SENT vs PARA). However, there are still many issues with PARA's translations. From the annotations and translators' comments, we observe that PARA suffers from occasional omissions of content from the source paragraph to a greater extent than SENT and GTR (see §D). Moreover, PARA still makes a sizeable number of mistranslations and grammatical errors, though fewer than SENT or GTR. These issues seem to be only partially mitigated by employing GPT-4 (see §F). Finally, it is important to acknowledge that the languages covered in the current study are either mid or high-resource. Performance might be much worse when translating from or into a low-resource language such as Zulu or Armenian.[27]

## Ethical considerations

**Translating with LLMs:** The rise of large language models has also brought many ethical concerns to the forefront of NLP research (Blodgett et al., 2020; Bender et al., 2021). LLMs encode biases and exhibit toxicity, and these behaviors can be exacerbated by unconstrained prompting (Gehman et al., 2020; Costa-jussà et al., 2022). Further ethical concerns arise in the context of machine translation, particularly *literary* translation, where multiple stakeholders – the author, the translator, and the audience – are involved (Taivalkoski-Shilov, 2019a). Low-quality output can influence the perception of the author's work, impair the reader's linguistic abilities, and hinder the transfer of ideas to the target language, while overrelying on machine translation can possibly threaten the role of human translators (Drugan, 2013; Ning and Domínguez, 2016; Taivalkoski-Shilov, 2019a). On the other hand, machine translation employed *responsibly* as an auxiliary tool holds the potential to alleviate the translator's cognitive burden (O'Brien, 2012) and make the author's work accessible to a broader audience more swiftly (Besacier, 2014). Contrary to the predictions in Eloundou et al. (2023), we do not view large language models as a substitute for human translators, but rather as a means to assist translators in their work.

**Human Evaluation:** The experiments involving human translators were reviewed by the IRB, and all involved translators gave their written consent to disclose their annotations, comments, and preference choices. In recognizing contributions, our acknowledgments only include the names of those translators who explicitly gave their consent to be acknowledged by their full name in this publication.

**Data Copyrights:** We use and make public only about 2% of the text from each of the original novels. This number was determined after consulting domain experts at the HathiTrust (https://www.hathitrust.org/) and qualifies as fair use (up to 10% of a text can generally be considered fair use).

## Acknowledgements

First and foremost, we would like to express our gratitude to the translators hired mostly on Upwork: Malgorzata Szymczak (*fr-pl*), Kinga Przekota (*ru-pl*), Michal Sikora (*cs-pl*), Paula Kurzawska (*de-pl*, *de-en*, *pl-en*), Kristy Darling Finder (*fr-en*), Timothy Shostak (*ja-en*), Shun Enoki (*zh-ja*), Takanori Kurokawa (*fr-ja*), Yoshiko Kikawa (*en-ja*), Shinnosuke Kasahara (*ru-ja*), and all those who wish to remain anonymous. We encourage any machine translation researchers working on these language pairs to contact these translators for human evaluations.

We would also like to show our appreciation to Jan Wislicki, Tom Gally, Nader Akoury, Kalpesh Krishna, Simeng Sun, Katherine Thai, and the entire UMass NLP group for insightful discussion, which helped to shape this project.

Furthermore, we would like to express our gratitude to the reviewers for their constructive feedback and valuable suggestions.

Finally, we would like to thank Sergiusz Rzepkowski (*pl*), Paula Kurzawska (*pl, en*), Hiroshi Iida (*ja*), Grégory Fleurot (*fr*), Peyton Bowman (*en*), Simeng Sun (*zh*), Igor Zapala (*pl, de*), Marvin Hoffmann (*de*), Kinga Przekota (*pl, ru*), and Yuki Mori (*ja*) for further consultations on their respective native languages.

This project was partially supported by awards IIS-1955567 and IIS-2046248 from the National Science Foundation (NSF) as well as an award from Open Philanthropy.

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

# Appendix

## A  Background

In this section of the appendix, we survey the existing approaches to document-level machine translation, which do not involve prompting LLMs.

**Existing approaches to document-level translation:**  Before the rise of neural machine translation, several attempts were made to incorporate discourse-level phenomena into statistical machine translation systems (Hardmeier, 2012; Carpuat and Simard, 2012; Hardmeier et al., 2013; Ding et al., 2014). Neural MT systems condition sentence-by-sentence translation on discourse-level context via concatenation models (Tiedemann and Scherrer, 2017; Jean et al., 2017; Agrawal et al., 2018; Junczys-Dowmunt, 2019; Lopes et al., 2020), hierarchical models (Miculicich et al., 2018; Tan et al., 2019; Chen et al., 2020; Zheng et al., 2020), multi-pass models (Mansimov et al., 2021), dynamic context models (Kang et al., 2020), multi-source models (Zhang et al., 2018; Feng et al., 2022), and transfer learning approaches (Zhang et al., 2022). Despite sometimes obtaining clear gains from discourse-level context (Voita et al., 2019), the machine translation community has not made much progress on this problem, particularly for non-English language pairs, due largely to the scarcity of parallel document-level corpora (Zhang et al., 2022). This problem has been partially addressed by introducing a pivot language (Cohn and Lapata, 2007; Utiyama and Isahara, 2007), but this approach can also lead to substantial information loss.

## B  Dataset

In this section of the appendix, we first discuss the rationale for the source and target language selection. Then we provide more details on the selection of the paragraphs. Finally, we provide details about the number of tokens and sentences in the source text and different translations.

**Target language selection:**  We select English, Japanese, and Polish as the target languages of our study, as these languages differ considerably in many linguistic aspects. English is an analytic language that is widely spoken and extensively studied in the field of natural language processing, and it serves as the primary pretraining language of most large language models, including GPT-3.5.[28] In contrast, both Japanese and Polish are comparatively under-explored. Japanese is an agglutinative language that employs three distinct writing systems: Kanji, Hiragana, and Katakana. As a high-context language, the translation of Japanese texts necessitates a profound comprehension of context and cultural nuances, rendering it a compelling choice for testing the limits of LLMs' translation capabilities. Polish, on the other hand, is a fusional language characterized by a rich morphological system. Its complex word forms, grammatical gender, conjugation, and declension make it an apt choice for testing the accuracy and robustness of LLMs.[29]

**Source language selection:**  As source languages, we select English (*es*), Polish (*pl*), Russian (*ru*), Czech (*cs*), French (*fr*), German (*de*), Japanese (*ja*), and Chinese (*zh*). These languages belong to a diverse array of language families – Indo-European (Romance, Germanic, Slavic), Sino-Tibetan, and Japonic – each with distinctive morphological traits – fusional, agglutinative, and analytic. Moreover, they employ a variety of writing systems such as the Latin alphabet, the Cyrillic alphabet, Hanzi, and Kanji/Hiragana/Katakana (see Table 4 for details). Finally, we carefully select source-target language pairs to ensure that our study encompasses both linguistically similar and dissimilar languages. For example, we paired *cs-pl*, as these languages are characterized by only 10% lexical distance[30] and have similar syntactic structures (Jágrová and Avgustinova, 2023). Conversely, we also include *ja-pl*, as the two languages have very little lexical overlap, vastly different grammars, and utilize distinct writing systems.

**Choosing paragraphs:**  The selection of a particular paragraph was semi-random, with certain considerations in mind during the sampling process. We prioritized the following criteria: (1) for each

---

[28]As of 2020, the reported distribution of languages featured in the present study within the GPT-3 training data was as follows: English – 92.647% (1st), French – 1.818% (2nd), German – 1.469% (3rd), Russian – 0.188% (9th), Polish – 0.155% (11th), Japanese – 0.111% (15th), Chinese – 0.099% (17th), Czech – 0.071% (18th) (see https://github.com/openai/gpt-3/blob/master/dataset_statistics/languages_by_word_count.csv). The current GPT-3.5 text-davinci-003 model is reported to incorporate data up to June 2021 and it is unclear what texts or languages were added to the original training data https://platform.openai.com/docs/models/gpt-3-5.

[29]The first author is fluent in all three target languages.

[30]i.e., the percentage of non-cognates in the language pair.

source language we sample paragraphs so that there is a combination of dialogue and narrative texts; (2) the paragraph should be reasonably intelligible to a human translator without additional context; and (3) alignment between the source paragraph and human translation should be feasible, meaning no major content rearrangement *across* paragraphs.

Nonetheless, meeting all these requirements was not always possible. For instance, the source text of *Convenience Store Woman* (*ja*) is mostly written in the first-person narrative. Since Japanese does not encode the speaker's gender in the verb forms, it is often impossible to determine whether the narrator is a male or a female. In cases where it was impossible to determine the gender of the character we instructed translators to accept *either* option, provided that the translation remained consistent within the given paragraph (i.e., the gender did not change within the paragraph).

**Dataset statistics:** The dataset used for this study contains 360 source paragraphs with their corresponding human translations.[31] We further report the following statistics: (1) the number of sentences in the source text, as per manual sentence tokenization, along with the number of tokens in the sources and text and each translation (Table 5), (2) the number of sentences in the source text, human translation, and each machine translation as tokenized with SPACY (Table 6), and (3) the distribution of sentences in paragraphs (Figure 7).

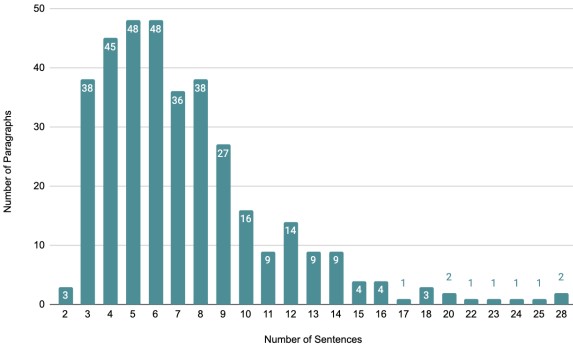

Figure 7: Distribution of sentences in the sampled paragraphs. The paragraphs were sentencized manually.

## C Prompting for Translation

## D Human Evaluation

In this section, we provide some further details about the human evaluation with a focus on the error annotation. First, discuss the issue of subjectivity in error annotation. Next, we explain some choices we had to make when annotating "inconsistency" and "format" errors. Finally, we present some details about the translators hired for the evaluation task.

**Error annotation:** Annotating and classifying errors in translations is inherently subjective (Lommel et al., 2014a; Han, 2020). For instance, translating French "corsage" ("bodice") as a "blouse" can be seen as either a mistranslation or a permissible deviation from the original text; this is, in fact, how the "corsage" was translated by the human translator in our data.

Furthermore, sometimes there are multiple ways of annotating errors (Thomson et al., 2023). Consider the following example:

(1) We had to hide the running, though, in case our haste betrayed us, so truer to say we slipped out quietly. When one of my parents appeared, my technique was: pretend to catch sight of someone in the next room. **Move in a natural manner toward this figment of my imagination, making a purposeful face.**

—ENGLISH SOURCE (from *A Children's Bible*)

The translation of the last sentence in (1) into Polish as an imperative can be considered a mistranslation. We would hypothesis that the system misinterpreted the source as an imperative form. However, using the infinitive form of the verb in the translation is less clear and raises questions about whether it is a mistranslation or a grammatical error. The distinction between the two lies in the point at which the mistake was made. If the original sentence was understood correctly but the resulting translation was ungrammatical, then it is a grammatical error. On the other hand, if the use of the infinitive form resulted from interpreting "move" as an infinitive, it may be considered a mistranslation as well.

**Inconsistency:** For marking the "inconsistency" errors we decided to the take *minimal* approach. For instance, is the same person is referred to in the translation as both "Piotr" and "Peter" we would

[31]See Table 7 for the list of novels included in the dataset and Table 8 for examples of aligned paragraphs.

| Language | Language Family | Morphological Features | Writing System |
|---|---|---|---|
| ENGLISH | Indo-European (Germanic) | Analytic | Latin Alphabet |
| GERMAN | Indo-European (Germanic) | Fusional | Latin Alphabet |
| FRENCH | Indo-European (Romance) | Fusional | Latin Alphabet |
| POLISH | Indo-European (Slavic) | Fusional | Latin Alphabet |
| CZECH | Indo-European (Slavic) | Fusional | Latin Alphabet |
| RUSSIAN | Indo-European (Slavic) | Fusional | Cyrillic |
| JAPANESE | Japonic | Agglutinative | Kanji / Hiragana / Katakana |
| CHINESE | Sino-Tibetan | Analytic | Hanzi |

Table 4: Details on languages included for the current study.

| LANG | #SENT | SRC | HUM | PARA | SENT | PARA_SENT | GTR |
|---|---|---|---|---|---|---|---|
| cs-pl | 163 | 2,154 | 2,027 | 2,122 | 2,123 | 2,259 | 2,065 |
| de-pl | 153 | 3,172 | 2,997 | 2,785 | 2,899 | 2,835 | 2,764 |
| ru-pl | 170 | 2,350 | 2,471 | 2,467 | 2,463 | 2,458 | 2,375 |
| ja-pl | 111 | 2,627 | 1,855 | 1,782 | 1,907 | 1,830 | 1,800 |
| en-pl | 127 | 1,702 | 1,526 | 1,444 | 1,513 | 1,483 | 1,462 |
| fr-pl | 119 | 3,253 | 2,789 | 2,641 | 2,673 | 2,654 | 2,543 |
| de-ja | 75 | 3,530 | 5,329 | 4,807 | 5,116 | 4,652 | 4,703 |
| en-ja | 176 | 1,959 | 2,617 | 2,538 | 2,653 | 2,617 | 2,634 |
| zh-ja | 194 | 2,998 | 4,124 | 3,861 | 4,249 | 3,957 | 3,978 |
| ru-ja | 193 | 2,539 | 4,753 | 3,982 | 4,348 | 4,088 | 3,921 |
| fr-ja | 195 | 2,510 | 3,426 | 3,110 | 3,355 | 3,106 | 2,958 |
| pl-ja | 188 | 1,953 | 2,944 | 3,083 | 3,418 | 3,199 | 2,972 |
| ja-en | 111 | 2,622 | 2,293 | 2,062 | 2,322 | 2,257 | 2,140 |
| pl-en | 148 | 2,696 | 3,430 | 3,234 | 3,290 | 3,273 | 3,213 |
| ru-en | 117 | 1,693 | 2,008 | 2,029 | 2,056 | 2,028 | 2,019 |
| fr-en | 120 | 3,253 | 3,123 | 3,067 | 3,150 | 3,064 | 3,098 |
| de-en | 153 | 3,172 | 3,346 | 3,361 | 3,413 | 3,325 | 3,314 |
| zh-en | 127 | 2,235 | 2,002 | 2,427 | 2,396 | 2,351 | 2,360 |
| **Total** | **2,640** | **46,418** | **53,060** | **50,802** | **53,344** | **51,436** | **50,319** |

Table 5: Number of sentences in the source text sentencized manually (#SENT) along with the number of tokens in the human reference (HUM) and different machine translations (PARA, SENT, PARA_SENT, GTR). All translations were tokenized using SPACY[32] with the large model for each of the three target languages (Polish, Japanese, and English). All source texts were tokenized with STANZA (Qi et al., 2020) as SPACY does not include models for all target languages.

Original text in Japanese:
「そういうのは実際には起こらないの? 」

Translation into Polish:
– To się w rzeczywistości nie zdarza?

(...)

Original text in Japanese:
「いらっしゃいませ! 」

Translation into Polish:

Figure 8: An example of prompt for SENT translations with one demonstration and a text to translate.

| Lang | Source | Target | Para | Sent | Para_Sent | GTr |
|------|--------|--------|------|------|-----------|-----|
| cs-pl | 168 | 177 | 167 | 169 | 181 | 168 |
| de-en | 155 | 182 | 166 | 167 | 164 | 155 |
| de-ja | 69 | 133 | 135 | 121 | 117 | 132 |
| de-pl | 155 | 170 | 166 | 167 | 169 | 157 |
| en-ja | 169 | 168 | 166 | 161 | 169 | 169 |
| en-pl | 131 | 127 | 130 | 132 | 130 | 131 |
| fr-en | 122 | 138 | 126 | 122 | 124 | 123 |
| fr-ja | 193 | 199 | 207 | 220 | 185 | 201 |
| fr-pl | 122 | 125 | 125 | 125 | 126 | 123 |
| ja-en | 101 | 120 | 116 | 116 | 116 | 111 |
| ja-pl | 101 | 127 | 117 | 115 | 118 | 108 |
| pl-en | 148 | 156 | 149 | 145 | 151 | 145 |
| pl-ja | 189 | 153 | 174 | 196 | 178 | 191 |
| ru-en | 123 | 119 | 121 | 124 | 121 | 123 |
| ru-ja | 144 | 155 | 158 | 161 | 164 | 196 |
| ru-pl | 168 | 172 | 170 | 171 | 172 | 172 |
| zh-en | 127 | 130 | 146 | 141 | 140 | 135 |
| zh-ja | 195 | 234 | 225 | 229 | 215 | 202 |
| **Total** | **2,580** | **2,785** | **2,764** | **2,782** | **2,740** | **2,742** |

Table 6: Number of sentences in the source text and each translation. The data was sentencized with SPACY. As evident from the data and manual inspection of translations the translations may result in a very different number of sentences as a result of splits and merges. We observe that about 55% of the data potentially lacks one-to-one correspondence.

Original text in Czech:
V hospodě U kalicha seděl jen jeden host. Byl to civilní strážník Bretschneider, stojící ve službách státní policie. **<translate>**Hostinský Palivec myl tácky a Bretschneider se marně snažil navázat s ním vážný rozhovor.**</translate>**

Translation into Polish:
W gospodzie „Pod Kielichem" siedział tylko jeden gość. Był to wywiadowca Bretschneider, będący na służbie policji państwowej. **<translated>**Gospodarz Palivec zmywał podstawki, a Bretschneider daremnie usiłował wyciągnąć go na poważną rozmowę.**</translated>**

(...)

Original text in Czech:
„Je to fakt dobrý," řekl mi Frodo, když se na můj výkres koukal. **<translate>**A skoro to vypadalo, že je z toho obrázku překvapenej.**</translate>** Pak se ptal, jestli maluju i doma, tak jsem odpověděla, že jo. Nejradši bych mu řekla i to, že chci být malířkou, ale mamka pořád opakuje, že je to blbost, že člověk musí dělat něco pořádnýho, jako třeba ona dělá knihovnici v knihovně, tak jsem raději mlčela, aby si nemyslel, že mám blbý nápady. Chvíli na výkres ještě koukal a otáčel ho ze všech stran a pak šel dál a koukal na obrázek Lindy, která nakreslila takový docela pěkný jabko.

Translation into Polish:
– To jest naprawdę dobre – powiedział Frodo, patrząc na moją pracę. **<translated>**

Figure 9: An example of prompt for PARA_SENT translations with one demonstration and a text to translate.

mark only the one that is less frequent. If "Piotr" appears *once* in the paragraph, while "Peter" is used *twice*, "Piotr" would be annotated as being inconsistent. The same strategy was applied for "register" errors, such as when both polite and casual forms were acceptable, but the translation used them randomly.

**Format:** We did not label "format" errors for the SENT and PARA_SENT translations, as we manually corrected the quotation marks during post-processing of the translations. This manual correction was done to ensure that SENT and PARA_SENT could be compared to PARA without relying too heavily on simple heuristic (i.e., incorrect usage of the quotation marks).

**Translators:** The translators in this study were hired on a freelancing platform, Upwork. We interviewed all translators prior to the task to assure

| Book title | Author | Translator(s) | Language | | Year Published | |
| --- | --- | --- | --- | --- | --- | --- |
| | | | Source | Target | Translation | Original |
| **A Children's Bible** | Lydia Millet | Aga Zano | *en* | *pl* | 2022 | 2020 |
| **What Can You See From Here** | Mariana Leky | Agnieszka Walczy | *de* | *pl* | 2021 | 2017 |
| **The Years** | Annie Ernaux | Krzysztof Jarosz & | *fr* | *pl* | 2022 | 2008 |
| | | Magdalena Budzińska | | | | |
| **Manaraga** | Wladimir Sorokin | Agnieszka Lubomira Piotrowska | *ru* | *pl* | 2018 | 2017 |
| **Crows** | Petra Dvorakova | Mirosław Śmigielski | *cs* | *pl* | 2020 | 2020 |
| **Convenience Store Woman** | Sayaka Murata | Dariusz Latoś | *ja* | *pl* | 2019 | 2016 |
| **Sixteen Horses** | Greg Buchanan | Fuji Yoshiko | *en* | *ja* | 2022 | 2021 |
| **An Inventory of Losses** | Judith Schalansky | Naoko Hosoi | *de* | *ja* | 2022 | 2018 |
| **Dear Reader** | Paul Fournel | Kei Takahashi | *fr* | *ja* | 2022 | 2011 |
| **The Shooting Party** | Anton Chekhov | Takuya Hara | *ru* | *ja* | 2022 | 1884 |
| **Sword of Destiny** | Andrzej Sapkowski | Yasuko Kawano | *pl* | *ja* | 2022 | 1992 |
| **Bare burial** | Fang Fang | Shin'ichi Watanabe | *zh* | *ja* | 2022 | 2016 |
| **What Can You See From Here** | Mariana Leky | Tess Lewis | *de* | *en* | 2021 | 2017 |
| **The Years** | Annie Ernaux | Alison L. Strayer | *fr* | *en* | 2017 | 2008 |
| **The Story of a Life** | Konstantin Paustovsky | Douglas Smith | *ru* | *en* | 2022 | 1956 |
| **The Books of Jacob** | Olga Yokarczuk | Jennifer Croft | *pl* | *en* | 2022 | 2014 |
| **Convenience Store Woman** | Sayaka Murata | Ginny Tapley Takemori | *ja* | *en* | 2018 | 2016 |
| **Cocoon** | Zhang Yueran | Jeremy Tiang | *zh* | *en* | 2022 | 2018 |

Table 7: Details of the translated novels used in our study. In cases where the same novel is used for multiple target languages (e.g., "The Years"), identical source paragraphs are extracted to enable comparisons across language pairs. These novels exhibit distinct differences beyond just their source languages. For instance, "What Can You See From Here" presents a philosophical exploration of life and death, while "Sword of Destiny" is a fantasy story part of "The Witcher" saga.

Original text in Japanese:
直子は立ちどまった。僕も立ちどまった。彼女は両手を僕の肩にあてて正面から、僕の目をじっとのぞきこんだ。彼女の瞳の奥の方ではまっ黒な重い液体が不思議な図形の渦を描いていた。そんな一対の美しい瞳が長いあいだ僕の中をのぞきこんでいた。それから彼女は背のびをして僕の頬にそっと頬をつけた。それは一瞬胸がつまってしまうくらいあたたかくて素敵な仕草だった。

Translation into Polish:
Naoko zatrzymała się. Ja też stanąłem. Położyła mi ręce na ramionach i zajrzała uważnie w oczy. W jej ciemnych jak atrament źrenicach tworzyły się przedziwne wirujące wzory. Te piękne oczy długo badały moje serce. Potem wyprostowała się i przytuliła policzek do mojego. To był cudowny ciepły gest, aż mi serce na chwilę zamarło.

(...)

Original text in Japanese:
店長は30歳の男性で、常にきびきびとしている。口は悪いが働き者の、この店で8人目の店長だ。
2人目の店長はサボり癖があり、4人目の店長は真面目で掃除好きで、6人目の店長は癖のある人で嫌われ、夕勤が全員一気に辞めるというトラブルになった。8人目の店長は比較的アルバイトからも好かれ、自分が体を動かして働くタイプなので、見ていて気持ちがいい。7人目の店長は気弱すぎて夜勤になかなか注意ができずに店がぼろぼろになってしまったので、少し口が悪くてもこれくらいのほうが働きやすいと、8人目の店長を見ると思う。

Translation into Polish:

Figure 10: An example of prompt for PARA translations with one demonstration and a text to translate.

that they were qualified to evaluate the translations. All translators were highly proficient in the source language and most of them were native speakers of the target language with some being bilingual.[33]

Only one translator reported familiarity with the book, which translation she evaluated. All translators were instructed to evaluate each paragraph in isolation without relying on any prior knowledge about the book and to allow for *all* possible interpretations based on the given part of the source text. They were asked to evaluate five translations first and received feedback on their work before moving forward. Details about the translators are reported

[33]We consider a translator bilingual only if they were raised using both languages; i.e. both can be consider their *native* languages (e.g., *ru-pl* translator was raised in Poland while speaking Russian at home). In the broader sense of this word, all of the translators are bilingual with some of them being trilingual. For the cases where the hired translator was *not* a native speaker of the target language, the annotations were verified by a native speaker of the target language in

consultation with the translator.

| Book | Lang Pair | Source | Target |
|------|-----------|--------|--------|
| An Inventory of Losses | de-ja | Natürlich hatte ich schon davor andere bemerkenswerte Begräbnisstätten besucht: die Toteninsel San Michele etwa, wie sie mit hohen, roten Backsteinmauern aus dem blaugrünen Wasser der Lagune von Venedig emporragt gleich einer uneinnehmbaren Festung, oder das grelle Jahrmarktstreiben des Hollywood Forever Cemetery am alljährlich vom mexikanischen Bevölkerung begangenen Día de los Muertos mit den orange-gelb geschmückten Gräbern und den von der fortgeschrittenen Verwesung auf ewig zum Grinsen verdammten Totenschädeln aus bunt gefärbtem Zucker und Pappmaché. Doch keine hat mich so berührt wie der Friedhof jener Fischersiedlung, in dessen eigentümlichem Grundriss — einer Art Kompromiss aus Kreis und Quadrat ich nichts anderes als ein Sinnbild der ungeheuerlichen Utopie zu erkennen glaubte, die ich dort verwirklicht sah: mit dem Tod vor Augen zu leben. Lange Zeit war ich überzeugt, an diesem Ort, dessen dänischer Name »kleine Insel« oder »vom Wasser umgeben« bedeutet, sei man dem Leben näher, gerade weil seine Bewohner die Toten wortwörtlich in ihre Mitte geholt hatten, anstatt sie wie sonst in unseren Breitengraden üblich — aus dem Innersten der Gemeinden vor die Stadttore zu verbannen, auch wenn der urbane Raum sich die Gräberstätten durch sein ungehemmtes Anwachsen oft nur wenig später wieder einverleibt hat. | もちろんそれ以前にもいくつか特筆すべき墓所を訪れたことはあった。たとえばヴェネツィアの干潟の青緑色の水中から、赤煉瓦の高い塀に囲まれて難攻不落の要塞のようにそびえたつ死者の島、サン・ミシェル。あるいはメキシコ系住民が毎年にぎやかに死者の日を祝う、ハリウッド・フォーエバー墓地。墓はオレンジと黄色の花で飾られ、カラフルな砂糖菓子や張り子細工の頭蓋骨は、腐敗が進んで永遠の笑顔を浮かべているようだ。けれども、この漁師町の墓地ほどに私の心を動かす墓所はなかった。まるで円と四角の間の妥協のようなその独特の輪郭に、私はまさにユートピアの象徴を見たように思った。死を目の前にしつつ生きるというユートピアが、そこに実現されていた。長いこと私は確信していた。デンマーク語で「小さな島」とか「水に囲まれた」という意味の名前を持つこの場所に住む人々は、同じくらいの緯度の国々で通常行われているように、共同体の内部から市門の外へと死者たちを追放する代わりに、死者たちを文字通り町の中心に迎え入れた。だからこそ、より生に近いのだと。もっとも都市空間もまた人口膨張のために、ほどなくして墓地をふたたび内部へと取り込まざるを得なくなるのだけれど。 |
| A Children's Bible | en-pl | The lady urinated. "Oh, poor old thing, she has a nervous bladder!" exclaimed someone's chubby mother. "Is that a Persian rug?" Whose mother was it? Unclear. No one would cop to it, of course. We canceled the performance. "Admit it, that was your mother," said a kid named Rafe to a kid named Sukey, when the parents had filed out. Some of their goblets, highball glasses, and beer bottles were completely empty. Drained. Those parents were in a hurry, then. "No way," said Sukey firmly, and shook her head. "Then who is your mother? The one with the big ass? Or the one with the clubfoot?" "Neither," said Sukey. "So fuck you." | Dama się posikała. – Och, biedactwo, ma wrażliwy pęcherz! – wykrzyknęła czyjaś pulchna matka. – Zaraz, to perski dywan? Czyją matką była? Nie wiadomo. Oczywiście nikt nie chciał się przyznać. Odwołaliśmy przedstawienie. – No dawaj, to twoja – powiedział chłopiec imieniem Rafe do dziewczynki imieniem Sukey, kiedy rodzice sobie poszli. Zostawili po sobie kieliszki, wysokie szklanki i butelki po piwie. Niektóre były zupełnie puste. Do ostatniej kropelki. Tym z rodziców się zatem spieszyło. – W życiu – odparła Sukey stanowczo i pokręciła głową. – To która? Ta z wielkim dupskiem? Czy ze szpotawą stopą? – Ani jedna, ani druga. Spierdalaj. |

Table 8: Examples of aligned reference source and target paragraphs from our dataset, including both a narrative (*An Inventory of Losses*) and a dialogue (*A Children's Biblie*). Our PARA approach takes as input the entire source paragraph and outputs a paragraph-level translation.

| Lang Pair | Title | Author | Translator(s) | Year Published Translation | Original |
|-----------|-------|--------|---------------|:---:|:---:|
| *ja-pl* | Norwegian Wood | Haruki Murakami | Dorota Marczewska & Anna Zielińska-Elliott | 1987 | 2006 |
| *de-pl* | The Trial | Franz Kafka | Jakub Ekier | 1925 | 2008 |
| *fr-pl* | Les Miserables | Victor Hugo | Krystyna Byczewska | 1862 | 1966 |
| *fr-pl* | The Little Prince | Antoine de Saint-Exupéry | Jan Szwykowski | 1862 | 1967 |
| *en-pl* | The Valley of Fear | Arthur Conan Doyle | Tadeusz Evert | 1915 | 1927 |
| *ru-pl* | War and Peace | Leo Tolstoy | Andrzej Stawar | 1869 | 1958 |
| *cs-pl* | War with Newts | Karel Čapek | Jadwiga Bułakowska | 1936 | 1949 |
| *pl-ja* | Solaris | Stanisław Lem | Mitsuyoshi Numano | 1961 | 2004 |
| *ru-ja* | Anna Karenina | Leo Tolstoy | Hakuyō Nakamura | 1878 | 2004 |
| *de-ja* | Der Steppenwolf | Hermann Hesse | Fujio Nagano | 1927 | 2000 |
| *fr-ja* | Around the World in 80 Days | Jules Verne | Yū Takano | 1873 | 2009 |
| *en-ja* | Animal Farm | George Orwell | Eitarō Sayama | 1945 | 1998 |
| *zh-ja* | Medicine | Lu Xun | Kōbai Inoue | 1919 | 1919 |
| *zh-ja* | The True Story of Ah Q | Lu Xun | Kōbai Inoue | 1921 | 1923 |
| *zh-ja* | Diary of a Madman | Lu Xun | Kōbai Inoue | 1921 | 1923 |
| *ru-en* | Confession | Leo Tolstoy | Peter Carson | 1882 | 2013 |
| *zh-en* | The Day the Sun Died | Yan Lianke | Carlos Rojas | 2015 | 2018 |
| *ja-en* | Kokoro | Natsume Sōseki | Edwin McClelan | 1914 | 1957 |
| *ja-en* | Kokoro | Natsume Sōseki | Meredith McKinney | 1914 | 2010 |
| *de-en* | Venus in Furs | Ritter von Leopold Sacher-Masoch | Fernanda Savage | 1870 | *unclear* |
| *fr-en* | The Debacle | Émile Zola | Leonard Tancock | 1870 | 1972 |

Table 9: List of novels employed in the prompts.

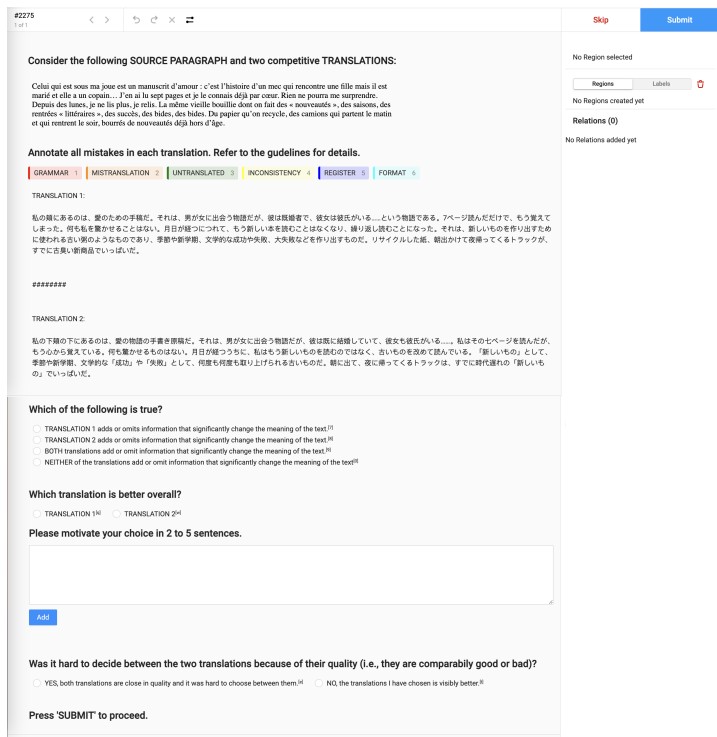

Figure 11: The annotation interface used for the error annotation task.

| LANG PAIR | NATIVE LANG | BOOK FAMILIARITY | GENDER |
|---|---|---|---|
| *zh-en* | Chinese | ✗ | Male |
| *ja-en* | English | ✗ | Male |
| *de-en* | Polish/English | ✗ | Female |
| *fr-en* | English | ✗ | Female |
| *ru-en* | Russian | ✗ | Female |
| *pl-en* | Polish/English | ✗ | Female |
| *en-ja* | Japanese | ✗ | Female |
| *fr-ja* | Japanese | ✗ | Male |
| *de-ja* | Japanese | ✗ | Female |
| *pl-ja* | Polish (author) | ✗ | Female |
| *ru-ja* | Japanese | ✗ | Male |
| *zh-ja* | Japanese | ✗ | Male |
| *de-pl* | Polish/English | ✗ | Female |
| *en-pl* | Polish (author) | ✗ | Female |
| *ru-pl* | Polish/Russian | ✗ | Female |
| *cs-pl* | Czech | ✗ | Male |
| *ja-pl* | Polish (author) | ✗ | Female |
| *fr-pl* | Polish | ✓ | Female |

Table 10: Details about the translators hired for the current annotation study. We note whether the translator was familiar with the source text prior to the evaluation task (*Book Familiarity*).

in Table 10.[34]

# E  Results

In this section of the appendix, we provide more detailed analysis of the results of the human evalu-

ation. We start with providing more details about the GTR vs PARA evaluation. Next, we include an in-depth discussion of the context-related errors in SENT which were corrected in the PARA translations. Finally, we include some comments from the translators. In the next section (§F), we also provide more information about the issues still present in the PARA translations along with the preliminary analysis of paragraph-level translation by GPT-4.

## E.1  PARA is clearly better than GTR

PARA translations are overwhelmingly preferred over those from Google Translate (GTR), with an 82.8% preference rate ($p<.001$, 95% CI [0.765, 0.880]). Even after removing the "unsure" votes, the preference for PARA remains significant at 88.0% ($p<.001$, 95% CI [0.812, 0.930]). In the *fr-ja*, *pl-ja*, *zh-ja*, and *cs-pl* language pairs, PARA received *all* of the ten votes over GTR. Part of this advantage may be attributed to GTR sometimes using English as a pivot language, which can result in information loss. Our Czech translator observed that mistakes in GTR translations suggest the text was first translated into English.[35]

---

[34]Three language pairs (*pl-ja*, *en-pl*, *ja-pl*) were annotated by the first author of this paper.

[35]For the *cs-pl* language pair, we separately annotated mistranslations arising from pivot translation. These errors accounted for over 50% of all mistranslations in that language pair. The elimination of the need for parallel data may therefore be beneficial for translating between lower-resource

Overall, GTR translations result in 57.7% more mistranslations, 37.3% more grammatical errors, over twice as many inconsistency errors, and ten times more register errors (see Table 2). Additionally, GTR produced 125 format errors while PARA produced perfect outputs in this regard. Finally, it is worth noting that GTR left fewer words untranslated, though this is inflated by the fact that in one German text, the word "Bauer" ("farmer") was untranslated 14 times in the PARA translation.

## E.2 Context-related errors

Here we present examples of context-related issues present in SENT while correctly translated by PARA.[36]

**Pronouns:** Unsurprisingly, the absence of discourse context results in the incorrect translation of pronouns. Consider the following example, with English glosses of important words provided in [brackets]:

(2)  И ветер [*wind*] то начинал шуметь в голых деревьях, то замолкал, так же как и я прислушиваясь к течению ночи. Но он [*he*] не уходил, он [*he*] был здесь.

    —RUSSIAN SOURCE (from *The Story of a Life*)

 a. The wind would start to rustle in the bare trees and then fall silent, just as I listened to the flow of the night. But he didn't leave, he was here.

    —GPT-3.5 SENT (ENGLISH)

 b. The wind would start to rustle in the bare trees, then die down, just like me, listening to the flow of the night. But it didn't go away, it was still here.

    —GPT-3.5 PARA (ENGLISH)

In Russian, nouns have grammatical gender. "Wind" in the first sentence of the source text is a masculine noun, so it is later referred to as "he" in (2). Without access to the context, the SENT model incorrectly translates it as "he" into English (2a), while the PARA translation correctly modifies the pronoun to "it" (2b).

When translating from Russian into Polish, another language with grammatical gender, we observe issues when the gender of Russian and Polish nouns differs. Consider the following example:

(3)  Романы, как известно, печатались на разной бумаге [*paper*]. И гореть она [*she*] может по-разному.

    —RUSSIAN SOURCE (from *Manaraga*)

 a. Romany, jak wiadomo, drukowano na różnym papierze [*paper*]. I może ona [*she*] tęsknić na różne sposoby.

    —GPT-3.5 SENT (POLISH)

 b. Jak wiadomo, powieści drukowano na różnym papierze [*paper*]. I może on [*he*] palić się na różne sposoby.

    —GPT-3.5 PARA (POLISH)

Although both Russian and Polish nouns possess grammatical gender, "Paper" in (3) is feminine in Russian and referred to as "she," whereas it is a masculine noun in Polish and should be referred to as "he," as in (3b). The absence of context in SENT leads to an incorrect translation in (3a).

**Cultural nuances:** Assigning appropriate pronouns without context becomes even more challenging when translating from languages like Japanese, in which speakers frequently refer to the listener (or themselves) in the third person rather than using second-person personal pronouns such as "you" in English. Consider the following example:

(4)  「気が付かなくてすみません」
「いやいや、(...)。古倉さんは毎日勤務 なのに手を抜かないからねー！」
[lit. *Ms./Mrs./Mr. Furukura works every day*]

    —JAPANESE SOURCE (from *Convenience Store Woman*)

 a. "I'm sorry I didn't notice."
"No, no, (...). Furukura-san works hard every day without taking any shortcuts!"

    —GPT-3.5 SENT (ENGLISH)

 b. "I'm sorry I didn't notice."
"No, no, (...). You work every day, but you never slack off!"

    —GPT-3.5 PARA (ENGLISH)

From the context of this conversation, a Japanese listener can easily infer that "Furukura-san" or "Miss Furukura"[37] in the last source sentence (4) is used instead of the second-person "you" as per Japanese convention. Translating this sentence without context into English, a language in which third-person reference is not common,[38] results in a confusing translation (4a) that implies that the speaker refers to some other "Furukura" rather than their listener. However, when translating the sentence in context, the model correctly changes "Furukura" into "you" (4b), which makes it clear whom the speaker refers to in English.

---

languages where sufficient parallel data is often unavailable necessitating the pivot translation.

[36]Note that PARA also suffers from context-related issues. However, at a much lesser extent than SENT.

[37]Note that the gender of neither character is apparent from the fragment alone.

[38]While third-person reference can be used in English, it is only used in rare circumstances e.g. when addressing children.

| Language Pair | Sent | Para | Para_Sent | Para | GTr | Para |
|---|---|---|---|---|---|---|
| *Russian - English* | 0 | 10 | 5 | 5 | 4 | 6 |
| *Chinese - English* | 1 | 9 | 3 | 7 | 3 | 7 |
| *Polish - English* | 4 | 6 | 4 | 6 | 1 | 9 |
| *French - English* | 5 | 5 | 4 | 6 | 2 | 8 |
| *Japanese - English* | 1 | 9 | 2 | 8 | 1 | 9 |
| *German - English* | 5 | 5 | 3 | 7 | 4 | 6 |
| Total | 16 | **44** | 21 | **39** | 15 | **45** |
| Percentage | 26.67% | **73.33%** | 35.00% | **65.00%** | 25.00% | **75.00%** |
| *German - Japanese* | 6 | 4 | 3 | 7 | 1 | 9 |
| *Russian - Japanese* | 4 | 6 | 4 | 6 | 2 | 8 |
| *French - Japanese* | 2 | 8 | 1 | 9 | 0 | 10 |
| *Polish - Japanese* | 2 | 8 | 4 | 6 | 0 | 10 |
| *English - Japanese* | 3 | 7 | 2 | 8 | 1 | 9 |
| *Chinese - Japanese* | 4 | 6 | 4 | 6 | 0 | 10 |
| Total | 21 | **39** | 18 | **42** | 4 | **56** |
| Percentage | 35.00% | **65.00%** | 30.00% | **70.00%** | 6.67% | **93.33%** |
| *English - Polish* | 0 | 10 | 3 | 7 | 4 | 6 |
| *Japanese - Polish* | 3 | 7 | 5 | 5 | 1 | 9 |
| *French - Polish* | 4 | 6 | 4 | 6 | 2 | 8 |
| *Czech - Polish* | 3 | 7 | 2 | 8 | 0 | 10 |
| *Russian - Polish* | 1 | 9 | 4 | 6 | 3 | 7 |
| *German - Polish* | 3 | 7 | 3 | 7 | 1 | 9 |
| Total | 14 | **46** | 21 | **39** | 11 | **49** |
| Percentage | 23.33% | **76.67%** | 35.00% | **65.00%** | 18.33% | **81.67%** |
| **TOTAL** | 51 | **129** | 60 | **120** | 30 | **150** |
| **PERCENTAGE** | 28.33% | **71.67%** | 33.33% | **66.67%** | 16.67% | **83.33%** |

Table 11: The number of votes for SENT vs PARA, PARA_SENT vs PARA, and GTR vs PARA in human evaluation by the language pair. The winning counts are highlighted in purple.

**Ellipsis:** Another example where context helps is the translation of elliptical constructions. Consider the following example:

(5)  „Ne, teď uděláš nádobí!" [*(you) will do the dishes!*]
„Neudělám!" [*(I) won't do!*]
„Uděláš!" [*(You) will do!*]
—Czech Source (from *Crows*)

  a.  — Nie, teraz zrobisz zmywanie! [*(you) will do the washing*]
— Nie zrobię! [*(I) won't do!*]
— Zrobisz to! [*(You) will do it!*]
—GPT-3.5 Sent (Polish)

  b.  — Nie, teraz umyjesz naczynia [*(You) will wash the dishes*]!
— Nie umyję [*(I) won't wash*]!
— Umyjesz [*(You) will wash*]!
—GPT-3.5 Para (Polish)

Czech uses the same collocation as English, "*do the dishes*" (5), which is invalid in Polish. Hence, the ellipses in the last two sentences in (5) require broader context to be translated correctly. PARA does it properly, translating both as "wash" (5b), while SENT unsurprisingly fails to choose the correct collocation (5a).

**Subject ellipsis:** Similarly, context may be needed to attribute a state or an action to the correct character due to the subject ellipsis. This is an obvious issue for languages like Japanese, which tend to omit the subject of the sentence and do not encode any relevant information in the verb form, but it can also arise in English. Consider the following example:

(6)  When we were done, the lipstick went back into some mother's Fendi handbag. We watched her apply it, unaware.
—English Source (from *A Children's Bible*)

  a.  Gdy skończyliśmy, szminka wróciła do jakiejś torebki Fendi należącej do matki. Patrzyliśmy, jak to robi, nieświadomi [*unaware (we)*] tego.
—GPT-3.5 Sent (Polish)

  b.  Kiedy skończyliśmy, szminka wróciła do torebki Fendi jakiejś matki. Patrzyliśmy, jak ją nakłada, nieświadoma [*unaware (she)*] naszych działań.
—GPT-3.5 Para (Polish)

From the second sentence alone it is not clear who is "unaware" (6) – the mother or the "we" (referring to children) watching her. Only from the

broader context can we confidently deduce that it is in fact the mother, not the children, who is "unaware." PARA (6b) correctly attributes the state of being "unaware" to the mother, which is exhibited by its usage of the singular feminine form of the adjective. In contrast, SENT (6a) mistranslates it using the plural masculine form of the adjective "unaware," which implies that it refers to "we" rather than the "mother."

**Consistency:** Context is sometimes critical for preserving the overall consistency of the text. The simplest cases include referring to the same entity – a place or a person – in the same way. More interesting cases pertain to style and can enhance the reader's experience. Consider the following example:

(7)  Alles zu vergessen, ist gewiss schlimm [*bad*]. Noch schlimmer [*worse*] ist, nichts zu vergessen (...).
—GERMAN SOURCE (from *An Inventory of Losses*)

a. すべて を 忘れる こと は 確かに悲惨な[*tragic*]ことです。さらに悪い[*worse*]のは、何も忘れないことです。
—GPT-3.5 SENT (JAPANESE)

b. すべてを忘れることは確かに悪い[*bad*]ことです。もっと悪い[*worse*]ことは、何も忘れないことです。
—GPT-3.5 PARA (JAPANESE)

The German source in (7) translates into English as "To forget everything is *bad*, certainly. *Worse* still is to forget nothing."[39] It is arguably important for the translation to repeat the same word which is an equivalent of the German "schlimm" ("bad"). PARA does it well, translating both as 悪い "warui," or "bad" (7b), in the exact same way as the human Japanese translator. SENT, on the other hand, uses two different words, "tragic" and "bad" (7a), which while technically correct omits the intentional repetition that is meant to introduce an unexpected conclusion.

**Polysemy:** The absence of context makes it difficult to interpret words or expressions that have multiple meanings in the source language. Consider the following example:

(8)  Все прошло хорошо. Книга прочитана идеально – не быстро и не медленно, минимум дыма. Классика. Я был в форме [*in shape*].
—RUSSIAN SOURCE (from *Maranaga*)

a. Wszystko poszło dobrze. Książka została przeczytana idealnie – nie szybko i nie wolno, minimalna ilość dymu. Klasyka. Byłem w mundurze [*in uniform*].

———
[39]Excerpt taken from the official English translation by Jakie Smith (2020).

—GPT-3.5 SENT (POLISH)

b. Wszystko poszło dobrze. Książka przeczytana idealnie – nie szybko i nie wolno, minimalna ilość dymu. Klasyka. Byłem w formie [*in shape*].
—GPT-3.5 PARA (POLISH)

The ambiguity stems here from multiple meanings of the Russian noun форма "forma" (8), which can mean either "shape" or "uniform." Since one can be "in shape" as well as "in a uniform", it is unclear from the sentence alone which meaning was intended by the author. From the preceding context, it is clear that "everything went well" for the narrator, who mastered the art of "book'n'grill," a unique form of expression exclusive to this fictional world. Based on this, we can infer that in this instance, the term "forma" signifies "shape," as in (8b), rather than "uniform," as in (8a).

**Appropriateness:** Finally, context may help to choose the more appropriate equivalent for the given situation. Consider the following example:

(9)  「あー、あと煙草の５番を一つ」
「かしこまりました」 [lit. *(I) understood*]
—JAPANESE SOURCE (from *Convenience Store Woman*)

a. "Ah, and one pack of cigarettes, number five." "Understood."
—GPT-3.5 SENT (ENGLISH)

b. "Ah, and one pack of cigarettes, number five." "Right away."
—GPT-3.5 PARA (ENGLISH)

The conversation above is between a clerk and a customer. The Japanese expression かしこまりました "kashikomarimashita" (9) is an honorific that literally means "understood." However, when choosing the best equivalent, the translator needs to consider the situation at hand to best reflect its meaning in the target language. "Understood" in SENT (9a) is technically correct, but it is an unfortunate word choice for the clerk to employ. On the other hand, "right away" in PARA (9b) fits much better in the context of this conversation. Had this been a series of commands (e.g., in a military context) "understood" would be the more favorable option.

### E.3 What do translators think about PARA?

To wrap up this section, we provide a qualitative analysis of the free-form comments written by translators to justify their preference judgments. Overall, the translators praise PARA for its *more skillful use of rhetoric devices*, and *surpas[ing]*

SENT *as a literary rendition.* They also mention that PARA *uses more of a poetic license but this makes it stylistically much smoother* than SENT. Furthermore, translators state that PARA *clearly better reflects the content and style of the original* when compared to GTR, and that it *stays consistent within the paragraph.* Inevitably, translations are not flawless, and there are instances where both compared systems fall short, as highlighted by one of the translators when assessing PARA against SENT: *Nightmare, a mistake upon mistake (...) Despite all these mistakes, I can understand the* [PARA] *translation better but they are equally miserable.*

## F  Limitations

In this section of the appendix, we delve deeper into the unresolved issues in the PARA translations. First, we discuss the omissions present in the translations. Next, we highlight some mistranslations that persist in the PARA translations. To conclude, we briefly discuss our initial experiments utilizing GPT-4 for paragraph-level translation.

**Omissions:**  One thing we ought to discuss is the omission issue. Upon examining translations and annotator feedback, we observe that PARA occasionally omits details, which are crucial to the storyline. Preliminary investigation indicates that PARA translations are more prone to omissions compared to SENT and GTR. Although PARA_SENT appears to mitigate this problem to some extent, it still results in a higher number of omissions than the sentence-level approach while at the same time introducing some repetition issues (see Table 12).[40]

**Mistranslations:**  Moreover, PARA still makes a sizeable number of mistranslations and grammatical errors, though fewer than SENT or GTR. We observe that PARA occasionally merges sentences with two distinctive subjects attributing all states and/or actions to one of them. Very rarely, we also find cases where context possibly confuses the model, resulting in an incorrect translation. The following example illustrates this issue:

---

[40]Note that although ask the annotators to report both omissions *and* additions, based on their comments and our analysis of the translations, we conclude that omissions are the predominant issue. In version two of our data (currently on `https://github.com/marzenakrp/LiteraryTranslation`), we further annotate any repetition as a separate type of error (i.e. error with 'repetition' label) rather than counting it as an addition. This annotations resulted in eight repetition errors in PARA_SENT translations.

(10)  Le bois du bureau amplifie les battements de mon cœur. Le vieux mobilier Art déco conduit bien les émotions et les fatigues. Ruhlman ? Leleu ? Il [*he*] en a tant vu.

—FRENCH SOURCE (from *Dear Reader*)

a.  机の木材が私の心臓の鼓動を増幅している。古いアール・デコ家具は感情や疲労をうまく導いてくれる。ルールマン？レルー？彼ら [*they*] はそんなに多くを見てきた。

—GPT-3.5 PARA (JAPANESE)

In the French text, the narrator wonders whether the brand of the desk was Ruhlman or Leleu, with both proper nouns possibly referring to a person. In the last sentence, the French text uses "il" or "he" (10), as a desk is a masculine noun in French ("le bureau"). PARA, on the other hand, appears to be confused by the two preceding names and incorrectly translates the singular pronoun as 彼ら, or "they."

Furthermore, we observe (very few) cases where the paragraph-level translation disregards the context. Most representative of this class of errors is when the model struggles to translate from Japanese in cases where the subject is omitted. The following example illustrates this issue:

(11)  ミホ [*Miho*] は、今では結婚して地元に中古の一戸建てを買っていて、そこに友達がよく集まっている。明日もアルバイトなので億劫に思う時もあるが、コンビニ以外の世界との唯一の接点であり、同い年の「普通の三十代女性」と交流する貴重な機会なので、ミホの [*Miho's*] 誘いにはなるべく応じるようにしている。

—JAPANESE SOURCE (from *Convenience Store Woman*)

a.  Miho [*Miho*] wyszła za mąż i kupiła stary, jednorodzinny dom w swoim rodzinnym mieście. Przychodzą tam często jej znajomi. Mimo że Miho ma [*Miho has*] jutro pracę w konbini, zazwyczaj chętnie odpowiada [*(she) responds*] na jej [*her*] zaproszenia, bo to jedyna okazja, by spotkać się z innymi kobietami w jej [*her*] wieku.

—GPT-3.5 PARA (POLISH)

b.  Miho is now married and has bought an old house in her hometown, where her friends often gather. Though she often finds it a chore to work tomorrow, it is her only connection to the world outside the convenience store, and a valuable opportunity to interact with other "normal thirty-something women" her age, so she tries to accept Miho's invitations as often as possible.

—GPT-3.5 PARA (ENGLISH)

Both Polish (11a) and English (11b) translations of the same source text (11) share a common issue. The narrator begins the paragraph by talking about Miho and then proceeds to describe her

| Language Pair | PARA | SENT | PARA_SENT | GTR |
|---|---|---|---|---|
| *Russian-English* | 0 | 0 | 1 | 0 |
| *Chinese-English* | 1 | 0 | 1 | 0 |
| *Polish-English* | 0 | 0 | 0 | 0 |
| *French-English* | 1 | 0 | 2 | 0 |
| *Japanese-English* | 2 | 1 | 2 | 3 |
| *German-English* | 0 | 0 | 0 | 0 |
| *German-Japanese* | 8 | 2 | 6 | 8 |
| *Russian-Japanese* | 10 | 4 | 6 | 4 |
| *French-Japanese* | 3 | 1 | 4 | 4 |
| *Polish-Japanese* | 4 | 1 | 3 | 0 |
| *English-Japanese* | 2 | 2 | 1 | 0 |
| *Chinese-Japanese* | 2 | 0 | 0 | 1 |
| *English-Polish* | 0 | 1 | 2 | 0 |
| *Japanese-Polish* | 0 | 0 | 1 | 1 |
| *French-Polish* | 2 | 2 | 1 | 1 |
| *Czech-Polish* | 1 | 2 | 1 | 0 |
| *Russian-Polish* | 1 | 1 | 1 | 0 |
| *German-Polish* | 0 | 0 | 0 | 0 |
| **Total** | **37** | 17 | 32 | 22 |

Table 12: Count of omissions reported by the translators for each translation method.

own (the narrator's) feelings about the situation, although the gender of the narrator is never revealed in the Japanese text. The second sentence should be written from a first-person perspective, particularly since it directly references Miho towards the end (blue text). However, both the Polish and English translations produced by PARA are confused by this: by using the third-person's perspective ("she," "her"), both translations incorrectly imply that Miho is the subject of the second sentence. SENT and GTR translate this passage accurately, albeit with some clumsy phrasing.

**GPT-4 does not magically solve all of these issues!** Our preliminary experiments indicate that GPT-4 (OpenAI, 2023) sometimes generates better paragraph-level translations than those of GPT-3.5. For instance, it seems to have a better grasp of the inverted word order in German, though no broader conclusions should be made without further testing. Nevertheless, it does not resolve all of the issues discussed in our paper. Mistranslations and grammatical errors are still abundant across many language pairs. GPT-4 produces the following translation when fed the previous example paragraph (11)

as input; note that all of the issues still remain:[41]

(12) Miho is now married and has bought a used single-family home in her hometown where her friends often gather. Although she sometimes finds it a drag to work a part-time job the next day, she makes an effort to respond to Miho's invitations because it's a valuable opportunity to interact with "normal" women in their thirties like herself, apart from her convenience store job.

—GPT-4 PARA (ENGLISH)

PARA translations hold the potential to captivate readers, especially if LLMs continue to improve at their current pace. Indeed, some of our translators mentioned that they genuinely enjoyed the task, though integrating these paragraphs into a coherent novel still poses a considerable challenge. With all that said, literary translation involves more than just overall "correctness" or mere entertainment value. A translation that is perfectly "correct" and enjoyable might still fail to convey the author's intentions or meaning skillfully hidden behind a simple phrase. Our *fr-en* translator shares her thoughts on this matter:

---

[41] Although the given paragraph is already comprehensible for a human reader, we also attempt to enhance the translation by incorporating three additional preceding paragraphs for context. Intriguingly, when provided with this extended context, both GPT-3.5 and GPT-4 generated accurate translations.

| SYSTEM | COMET | BLEURT | BERTSCORE | COMET-QE |
|---|---|---|---|---|
| PARA | **0.785** | **0.485** | **0.840** | **0.038** |
| SENT | 0.779 | 0.469 | 0.839 | -0.052 |
| PARA_SENT | 0.780 | 0.480 | 0.838 | -0.062 |
| GTR | 0.735 | 0.443 | 0.832 | -0.156 |

Table 13: Results of automatic evaluation. A higher number indicates better scores.

> *Both translations* [SENT and PARA] *translate the words without the feeling; the original author's voice is lost.*
>
> —FRENCH TO ENGLISH TRANSLATOR

## G Automatic Evaluation

In this section of the appendix, we present the results of automatic evaluation. First, we discuss the scores assigned to the translations by automatic metrics.[42] Then we provide the statistical analysis. Finally, we present the correlation of each metric with human judgments for the 180 paragraphs used in the human evaluation.

**Automatic metrics favor PARA:** We assess the translation from all four systems using the reference-based COMET (Rei et al., 2022), BLEURT (Sellam et al., 2020), and BERTSCORE (Zhang et al., 2020) metrics, as well as the reference-free COMET-QE (Rei et al., 2021)[43] metric. Although these metrics were not explicitly designed for evaluating paragraph-level outputs and their results should be interpreted with caution, they prove more reliable than string-based metrics like BLEU, especially for literary translations (Thai et al., 2022; Karpinska et al., 2022; Gehrmann et al., 2022). Table 13 shows the effectiveness of the PARA translation method: a statistical analysis with linear mixed-effects models (Baayen et al., 2008) demonstrates that PARA significantly outperforms SENT and GTR based on COMET, BLEURT, and COMET-QE scores ($p<.001$), and surpasses GTR based on the BERTSCORE results ($p<.001$). We discuss the details of this statistical analysis in the next section.

**Statistical Analysis:** We employ the linear-mixed effect models (Baayen et al., 2008) to analyze the scores produced by automatic metrics.

---

[42]This analysis is done on the entire dataset excluding only the paragraphs which were too long as per each metric's token limit.

[43]We use the newest `wmt22-comet-da` checkpoints for COMET, `Bleurt-20` checkpoints for BLEURT, `wmt20-comet-qe-da` checkpoints for COMET-QE, and the HuggingFace implementation which employs `roberta-large` for BERTSCORE.

| METRIC | ACC | $\tau$ | ACC (*conf*) | $\tau$ (*conf*) |
|---|---|---|---|---|
| COMET | 67.41% | 0.348 | 72.78% | 0.456 |
| COMET-QE | 64.44% | 0.289 | 70.64% | 0.413 |
| BLEURT | 61.30% | 0.226 | 66.36% | 0.327 |
| BARTSCORE | 58.52% | 0.170 | 63.91% | 0.278 |

Table 14: Correlation of automatic metrics with human judgments from our human evaluation. We evaluate the metrics performance on *all* human judgments as well as on the *subset* of judgments where the translator indicated that the chosen translation was visibly better (*conf*). We report both the percentage of agreement (ACC) and Kendall's Tau ($\tau$). Data reported on v1 of the dataset.

We fitted the model in R using the `lme4` package (Bates et al., 2015); the *p*-values were obtained with the `LmerTest` package (Kuznetsova et al., 2017). Linear-mixed effects models contain both *fixed-effects* and *random-effects* (random *intercept* and/or *slope*). The fixed effect here is the translation setup (PARA, SENT, PARA_SENT, GTR) with the source paragraph being coded as the random effect (random intercept). We inspect the residual plots to ensure that the variance across the fitted range is relatively constant. The results from the fitted model are presented in Table 18 (BLEURT), Table 20 (COMET), Table 22 (COMET-QE), and Table 24 (BERTSCORE).[44]

We further perform a post hoc analysis using the `emmeans` package (Lenth, 2023) to obtain *p*-values for the pairwise comparison. The results of the post hoc analysis are presented in Table 19 (BLEURT), Table 21 (COMET), Table 23 (COMET-QE), and Table 25 (BERTSCORE).

**Correlation with Human Judgements:** We investigate the correlation of automatic metrics with human judgments in our evaluation. We consider (1) all the judgments, as well as (2) a subset of all judgments where the annotator stated that they were sure that one translation is *clearly* better than the other. We compute both *accuracy* (i.e., the percentage of cases where the metric agrees with human judgment), and a correlation coefficient Kendall's Tau which is defined as follows:

$$\tau = \frac{\text{Concordant} - \text{Discordant}}{\text{Concordant} + \text{Discordant}}$$

---

[44]It should be noted that, while significant, the analysis is underpowered. It is possible that analyzing more examples would provide a more reliable analysis.

| Source | Target | Para | Para_Pivot |
|--------|--------|------|------------|
| *Czech* | *Polish* | 11 | 9 |
| *German* | *Japanese* | 13 | 7 |
| *German* | *Polish* | 12 | 8 |
| *French* | *Japanese* | 9 | 11 |
| *French* | *Polish* | 11 | 9 |
| *Japanese* | *Polish* | 10 | 10 |
| *Polish* | *Japanese* | 3 | 17 |
| *Russian* | *Japanese* | 10 | 10 |
| *Russian* | *Polish* | 8 | 12 |
| *Chinese* | *Japanese* | 9 | 11 |
| | Total | *96* | *104* |

Table 15: The results of pairwise comparison for the paragraph-level translations with (Para_Pivot) and without (Para) English as a pivot language.

Table 14 shows the correlation of automatic metrics with the human judgments obtained in this study. Comet exhibits the highest agreement with human judgments both in terms of the *accuracy* (64.04% for all data, 72.78% for confident votes only) and Kendall's Tau (0.341 for all data, 0.456 for confident votes only).

# H  Pivot Pilot

In this section of the appendix, we discuss the results of the preliminary study where we translated the paragraphs using English as a pivot language. Table 15 shows the results of this pilot study. The evaluation was done by the first author on *all* 20 passages for every language pair that did not include translation from or into English, as these do not require any pivoting.[45] A total number of 200 pairs was evaluated employing simple preference judgments.

During the Para_Pivot translation process, the model utilized both the source text and its corresponding English translation (`text-davinci-003`, top-p=1.0, temp=0.3). This approach has the potential to mitigate the limitations associated with pivoting translations, where some information may be lost. For example, both Czech and Polish encode the gender information in the past tense form of the

---

[45]The author is fluent in English, Japanese, and Polish with a limited proficiency in other source languages.

verb. English does not, so this information is lost and will most likely result in an erroneous translation. Indeed, we notice that adding the source text helps the model to overcome this shortcoming, however, we do not observe a clear gain from using English as a pivot language. Consider the following example:

(13) Taťka mě zase zmlátil. Byl hrozně naštvanej, protože mamka řvala. On mě vždycky mlátí, když ona řve. Že prej jsem pořádně neudělala nádobí. Ale já vím, že jsem je udělala dobře. I ten hrnec jsem vydrhla pořádně. A ještě jsem to všechno utřela před koupáním. Ale možná jsem nevyždímala hadr.
    —Czech Source (from *Crows*)

a. Tata mnie znowu zbił. **Był wściekły**, bo mama krzyczała. On zawsze mnie bije, kiedy ona krzyczy. Że niby nie **umyłam** dobrze naczyń. Ale ja wiem, że **umyłam** je dobrze. I garnek też dokładnie **wypolerowałam**. I jeszcze wszystko **wytrzepałam** przed kąpielą. Ale może nie **wycisnıłam** ręcznika.
    —GPT-3.5 Para (Polish)

b. Tata mnie znów uderzył. **Był wściekły**, bo mama krzyczała. On zawsze mnie bije, kiedy ona krzyczy. Że niby nie **umyłam** dobrze naczyń. Ale ja wiem, że **umyłam** je dobrze. Nawet garnek dokładnie **wytrzepałam**. I jeszcze wszystko przed kąpielą **wytrzepałam**. Ale może nie **wyżągnęłam** mopa.
    —GPT-3.5 Para_Pivot (Polish)

In each instance, the emphasized verbs could potentially be mistranslated when translated through English as the pivot language, as the speaker's gender information would be lost. For instance, the past tense verb "washed" remains unchanged in English regardless of the gender of the speaker, with such details encoded only in the source (*Czech*) and target (*Polish*) languages. In this case, all verbs have been translated accurately with respect to grammatical gender, implying that incorporating the source language into the pivot pipeline does indeed improve the translation. However, Para_Pivot still selects less suitable verbs (highlighted in red) resulting in slightly more errors in this particular paragraph.

The only pair where pivoting seems to help is *pl-ja*. While it is unclear why this happens, it is possible that this outcome is due to the specifics of the Polish novel employed for the translation. *Sword of Destiny* by Andrzej Sapkowski uses a very distinct language with many archaic expressions. It is possible that translating into English, a language the GPT models were trained on, helps the model deal with these difficult phrases.

Since we do not observe any apparent gains from performing the translation via English as a pivot

language ($p$=0.62, 95% [0.448, 0.591]) and doing so reduces the number of examples one can fit into the prompt, we continue our experiments with a direct translation.

| TYPE | DESCRIPTION | TRG LANG | PARA | SENT | PARA_SENT | GTR |
|------|-------------|----------|------|------|-----------|-----|
| CONTEXT (SENTENCE) | A mistranslation that results most likely from lack of "understanding" the sentence-level context (e.g., translating "guide" as "doradca," or "adviser" instead of "przewodnik," or "guide"). This can include translating a word or a phrase into one that is semantically related but does not convey the intended meaning, or translation which appear to be an outcome of translating a word semantically related to the source word, instead of the source word itself. | Japanese | 114 | 118 | 107 | 158 |
| | | Polish | 64 | 67 | 49 | 82 |
| | | English | 30 | 36 | 44 | 59 |
| CONTEXT (PARAGRAPH) | A mistranslation that results from lack of a beyond-sentence context. This include issues such as polysemy, employment of correct pronouns, or translating elliptical expressions. | Japanese | 6 | 36 | 6 | 38 |
| | | Polish | 13 | 51 | 15 | 59 |
| | | English | 2 | 25 | 0 | 48 |
| MINOR ISSUE | A minor issue which does not significantly affect the text and can be disputable, such as translating "barked" as "howl." | Japanese | 34 | 25 | 26 | 16 |
| | | Polish | 33 | 26 | 16 | 13 |
| | | English | 18 | 11 | 12 | 9 |
| SURFACE SIMILARITY | A translation by word which is similar to the correct translation on the surface level, but has a different meaning (e.g., "Wilczak," a Polish surname, instead of "wilczarz," a "wolfhound"). | Japanese | 8 | 6 | 7 | 2 |
| | | Polish | 14 | 13 | 16 | 5 |
| | | English | 5 | 5 | 6 | 2 |
| WORD-BY-WORD | A translation of longer phrase which is overly literal resulting in confusing and incorrect translation. | Japanese | 15 | 52 | 34 | 84 |
| | | Polish | 17 | 23 | 18 | 33 |
| | | English | 7 | 13 | 5 | 20 |
| UNRELATED WORD | A translation with unrelated word such as "klnie" ("swear") instead of "zapuka" ("knock") where no apparent semantic relation could be found. | Japanese | 3 | 2 | 5 | 4 |
| | | Polish | 5 | 14 | 10 | 12 |
| | | English | 1 | 3 | 1 | 2 |
| SUBJECT CHANGED | Change of subject. In the case of PARA, it occurs mostly due to merging two sentences with two distinctive subjects where all states and/or actions are then assigned to one of them. | Japanese | 5 | 2 | 2 | 0 |
| | | Polish | 6 | 0 | 5 | 3 |
| | | English | 7 | 2 | 5 | 1 |
| FACTUALITY | A translation that results in change in factuality, such as translating affirmative sentence as negation or translating word by its antonym. | Japanese | 4 | 11 | 5 | 7 |
| | | Polish | 0 | 2 | 1 | 3 |
| | | English | 1 | 2 | 1 | 1 |
| NON-WORD | A translation by a non-existent (made up) word. Some examples include skillfully constructed words like 火炎棒 which was generated instead of a "torch." While this word does not exist in Japanese (or Chinese) it follows the compositionality rules of these languages and is fully intelligible to a native speaker (火炎 "fire" and 棒 "stick.") | Japanese | 1 | 2 | 2 | 0 |
| | | Polish | 6 | 8 | 9 | 3 |
| | | English | 0 | 0 | 0 | 0 |
| MOOD | Change in the grammatical mood with regard to the source text. Note that the sentence here is *still* grammatically correct but does not reflect the meaning intended by the author. | Japanese | 4 | 9 | 1 | 3 |
| | | Polish | 1 | 3 | 4 | 2 |
| | | English | 0 | 0 | 0 | 0 |
| UNNECESSARY TRANSLATION | A translation of text which should be left untranslated such as some proper names. | Japanese | 0 | 0 | 0 | 0 |
| | | Polish | 0 | 3 | 0 | 2 |
| | | English | 1 | 1 | 1 | 1 |
| LANGUAGE MISMATCH | A translation into a language different than the target language (e.g., Chinese instead of Japanese). Note that leaving the word in the source language classifies as an "untranslated" error. | Japanese | 2 | 3 | 3 | 2 |
| | | Polish | 2 | 0 | 2 | 0 |
| | | English | 0 | 0 | 0 | 0 |
| NUMBER/TIME | A translation which changes number or time expression, such as translating 1h15min as 1h30min. Note that these rarely affect the overall meaning of the text. We have not observe cases where this would be a critical issue. | Japanese | 3 | 2 | 4 | 3 |
| | | Polish | 0 | 0 | 0 | 0 |
| | | English | 5 | 2 | 1 | 3 |
| PIVOT TRANSLATION (Czech) | A mistranslation that stems from pivoting on English (annotated for cs-pl language pair). | Polish | 0 | 0 | 0 | 43 |
| OTHER | Other issues which do not fit into any of the above. | Japanese | 24 | 26 | 27 | 17 |
| | | Polish | 9 | 14 | 10 | 13 |
| | | English | 10 | 4 | 5 | 4 |
| | | TOTAL (*Japanese*) | 223 | 294 | 229 | 334 |
| | | TOTAL (*Polish*) | 170 | 224 | 155 | 273 |
| | | TOTAL (*English*) | 87 | 104 | 81 | 150 |
| | | TOTAL (*All*) | **480** | **622** | **465** | **757** |

Table 16: Classification of mistranslation errors for each system grouped by the target language. The manual classification was performed on the v1 of the annotated dataset.

| Trg Lang | Type | SubType | Para | Sents | Para_Sents | GTr |
|---|---|---|---|---|---|---|
| Japanese | Particle | wrong or missing | 21 | 22 | 13 | 12 |
|  | Adjective | wrong continuative | 0 | 2 | 3 | 0 |
|  |  | other | 0 | 0 | 2 | 0 |
|  | Verb | tense | 3 | 7 | 1 | 14 |
|  |  | mood | 2 | 1 | 4 | 5 |
|  |  | finite/non-finite | 5 | 2 | 1 | 3 |
|  |  | other | 2 | 5 | 6 | 0 |
|  | Order | wrong order | 1 | 6 | 1 | 16 |
|  | Other |  | 8 | 5 | 6 | 13 |
|  | Total |  | 42 | 50 | 37 | 63 |
| Polish | Adjective | gender | 7 | 14 | 8 | 4 |
|  |  | case | 2 | 1 | 1 | 0 |
|  |  | other | 1 | 1 | 1 | 1 |
|  | Noun | case | 9 | 13 | 9 | 1 |
|  |  | other | 3 | 3 | 3 | 2 |
|  | Pronoun | omitted or wrong | 5 | 8 | 3 | 2 |
|  |  | case or gender | 1 | 6 | 4 | 5 |
|  | Verb | aspect | 1 | 5 | 1 | 12 |
|  |  | person or gender | 2 | 8 | 5 | 2 |
|  |  | conjugation | 1 | 0 | 7 | 3 |
|  |  | other | 2 | 4 | 1 | 13 |
|  | Preposition | omitted or wrong | 14 | 15 | 15 | 4 |
|  | Numeral | case or gender | 2 | 1 | 0 | 1 |
|  | Order | wrong order | 2 | 4 | 2 | 4 |
|  | Other |  | 3 | 3 | 4 | 5 |
|  | Total |  | 55 | 86 | 64 | 59 |
| English | Article | omitted or wrong | 1 | 9 | 2 | 8 |
|  | Preposition | omitted or wrong | 3 | 7 | 3 | 5 |
|  | Other |  | 1 | 4 | 4 | 5 |
|  | Total |  | 5 | 20 | 9 | 18 |

Table 17: Categorization of grammar errors in each translation configuration, grouped by the target language. The manual classification was performed on the v1 of the annotated dataset.

| | BLEURT | | |
|---|---|---|---|
| Predictors | Estimates | CI | $p$-value |
| (Intercept) | 0.48 | 0.47–0.50 | <**0.001** |
| Para_Sent | -0.00 | -0.01–0.00 | 0.130 |
| Sent | -0.02 | -0.02–(-0.01) | <**0.001** |
| GTr | -0.04 | -0.05–(-0.04) | <**0.001** |

Table 18: Results of linear-mixed effects models analysis for BLEURT scores.

|  | BLEURT | | | | |
| Contrast | Estimate | SE | df | *t*-ratio | *p*-value |
| --- | --- | --- | --- | --- | --- |
| PARA - PARA_SENT | 0.00477 | 0.00315 | 1074 | 1.515 | 0.780 |
| PARA - SENT | 0.01641 | 0.00315 | 1074 | 5.215 | **<0.001** |
| PARA - GTR | 0.04155 | 0.00315 | 1074 | 13.205 | **<0.001** |
| PARA_SENT - SENT | 0.01164 | 0.00315 | 1074 | 3.700 | **0.001** |
| PARA_SENT - GTR | 0.03678 | 0.00315 | 1074 | 11.690 | **<0.001** |
| SENT - GTR | 0.02514 | 0.00315 | 1074 | 7.990 | **<0.001** |

Table 19: Result of post hoc analysis with *emmeans* package for BLEURT.

|  | COMET | | |
| Predictors | Estimates | CI | *p*-value |
| --- | --- | --- | --- |
| (Intercept) | 0.79 | 0.77–0.80 | **<0.001** |
| PARA_SENT | -0.01 | -0.01–(-0.00) | **0.019** |
| SENT | -0.01 | -0.01–(-0.00) | **0.004** |
| GTR | -0.05 | -0.05–(-0.05) | **<0.001** |

Table 20: Results of linear-mixed effects models analysis for COMET scores.

|  | COMET | | | | |
| Contrast | Estimate | SE | df | *t*-ratio | *p*-value |
| --- | --- | --- | --- | --- | --- |
| PARA - PARA_SENT | 0.00563 | 0.00239 | 1074 | 2.356 | 0.112 |
| PARA - SENT | 0.00691 | 0.00239 | 1074 | 2.893 | **0.023** |
| PARA - GTR | 0.04998 | 0.00239 | 1074 | 20.928 | **<.001** |
| PARA_SENT - SENT | 0.00128 | 0.00239 | 1074 | 0.536 | 1.000 |
| PARA_SENT - GTR | 0.04435 | 0.00239 | 1074 | 18.571 | **<.001** |
| SENT - GTR | 0.04307 | 0.00239 | 1074 | 18.035 | **<.001** |

Table 21: Result of post hoc analysis with *emmeans* package for COMET.

|  | COMET-QE | | |
| Predictors | Estimates | CI | *p*-value |
| --- | --- | --- | --- |
| (Intercept) | -0.04 | -0.06 – -0.01 | **0.004** |
| PARA_SENT | -0.01 | -0.03 – -0.00 | **0.026** |
| SENT | -0.02 | -0.04 – -0.01 | **<0.001** |
| GTR | -0.12 | -0.13 – -0.11 | **<0.001** |

Table 22: Results of linear-mixed effects models analysis for COMET-QE scores.

|  | COMET-QE | | | | |
| Contrast | Estimate | SE | df | $t$-ratio | $p$-value |
| --- | --- | --- | --- | --- | --- |
| PARA - PARA_SENT | 0.01464 | 0.00655 | 1074 | 2.235 | 0.154 |
| PARA - SENT | 0.02376 | 0.00655 | 1074 | 3.628 | **0.002** |
| PARA - GTR | 0.11848 | 0.00655 | 1074 | 18.092 | **<.001** |
| PARA_SENT - SENT | 0.00912 | 0.00655 | 1074 | 1.392 | 0.9844 |
| PARA_SENT - GTR | 0.10384 | 0.00655 | 1074 | 15.857 | **<.001** |
| SENT - GTR | 0.09472 | 0.00655 | 1074 | 14.464 | **<.001** |

Table 23: Result of post hoc analysis with *emmeans* package for COMET-QE.

|  | BERTSCORE | | |
| Predictors | Estimates | CI | $p$-value |
| --- | --- | --- | --- |
| (Intercept) | 0.84 | 0.83–0.85 | <**0.001** |
| PARA_SENT | -0.00 | -0.00–0.00 | **0.037** |
| SENT | -0.00 | -0.00–0.00 | 0.522 |
| GTR | -0.01 | -0.01–0.01 | <**0.001** |

Table 24: Results of linear-mixed effects models analysis for BERTSCORE scores.

|  | BERTSCORE | | | | |
| Contrast | Estimate | SE | df | $t$-ratio | $p$-value |
| --- | --- | --- | --- | --- | --- |
| PARA - PARA_SENT | 0.002422 | 0.00116 | 1074 | 2.082 | 0.225 |
| PARA - SENT | 0.000745 | 0.00116 | 1074 | 0.640 | 1.000 |
| PARA - GTR | 0.007508 | 0.00116 | 1074 | 6.454 | <**0.001** |
| PARA_SENT - SENT | -0.001678 | 0.00116 | 1074 | -1.442 | 0.897 |
| PARA_SENT - GTR | 0.005086 | 0.00116 | 1074 | 4.372 | <**0.001** |
| SENT - GTR | 0.006763 | 0.00116 | 1074 | 5.814 | <**0.001** |

Table 25: Result of post hoc analysis with *emmeans* package for BERTSCORE.