# OpenReview forum: "Large language models effectively leverage document-level context for literary translation, but critical errors persist"
_EMNLP/2023/Conference — Submitted to EMNLP 2023_

### Official Review · Reviewer_2yxk · 2023-08-01

**Soundness:** 3

**Ethical Concerns:**

Yes

**Excitement:**

3: Ambivalent: It has merits (e.g., it reports state-of-the-art results, the idea is nice), but there are key weaknesses (e.g., it describes incremental work), and it can significantly benefit from another round of revision. However, I won't object to accepting it if my co-reviewers champion it.

**Justification For Ethical Concerns:**

I am not sure about the data practices in the paper as some of the data used seemed to be from behind a paywall but was not paid for (see Footnote 6).

**Missing References:**

https://aclanthology.org/2023.acl-long.435.pdf among others discuss evaluation of document-level translation

**Paper Topic And Main Contributions:**

This work examines the ability of GPT-3.5 to translate paragraph-level literary texts versus sentence-level literary text. Literary texts pose many unique challenges to translators. The authors collect a dataset of 360 aligned source-target paragraphs from after 2021 in 18 languages. They then solicit a human evaluation using the annotation protocol of
 Multidimensional Quality Metrics (MQM), with annotators identifying span-level errors in the translations and ranking the two sets of translations. The annotators make binary judgments as to whether they prefer GPT-3.5 translations prompted to perform sentence-level isolated translation, sentence-level translation with the paragraph as context, or paragraph-level translation. The results indicate that prompting for paragraph translation leads to the most preferred output and that translations into Japanese contain more mistakes, while GPT-3.5 often generates incorrect gender, case, or prepositions in Polish.

**Questions For The Authors:**

Question A: You state that the data will be made available. How will this be possible if some of the data is behind a paywall, i.e. in the cases where you took paragraphs from Amazon's free preview functionality?

Question B: Why did you choose the subset of MQM translation span-level errors that you did?

**Reasons To Accept:**

This work investigates the quality of GPT-3.5 translations if performed at the sentence or paragraph-level and with or without source paragraph context, which is an interesting question and seemingly novel. The human evaluation is fairly in depth, both exploring the errors made by GPT-3.5 as well as binary preference of one output versus another. The human evaluation also includes three language pairs and the literary domain enables interesting analyses given the creative voice necessary for effective literary translation. The results are clear and are well-explained. The data collection being only on works after 2021 also likely avoids problems with overlap with training data.

**Reasons To Reject:**

The authors use a checkpoint of GPT-3.5 which has been fine-tuned to follow instructions from human feedback, but it seems that the best model to test a discrepancy in GPT-3.5 translation quality at the document versus sentence-level would be a version that had been fine-tuned on literary texts.

The work claims there there are no reliable document-level translation metrics as part of the motivation for this work. Relevant work on evaluating document-level translation and the Background section is thus lacking, with even the Background in Appendix A does not address this claim.

The work consistently claims that the findings apply to LLMs, when in fact the findings only apply to GPT-3.5. This should be clarified and considerations of how GPT-3.5 is similar to and different from other LLMs would be important to include if discussing LLMs generally.

**Reproducibility:**

2: Would be hard pressed to reproduce the results. The contribution depends on data that are simply not available outside the author's institution or consortium; not enough details are provided.

**Reviewer Confidence:**

2: Willing to defend my evaluation, but it is fairly likely that I missed some details, didn't understand some central points, or can't be sure about the novelty of the work.

**Typos Grammar Style And Presentation Improvements:**

The authors conflate document-level and paragraph-level regularly throughout the paper; it would be more effective to choose and stick to one or to explain why use them in different places in the paper.

The number of hours required for annotation being stated in the abstract seems a little gauche and, if necessary to state, would be better suited for the main paper.

Many of the figures felt unnecessary (Figure 1, Table 1, Figure 4) or did not present the relevant information effectively (Table 2, Figure 6, Table 3) which contrasted with the exorbitant amount of information subjected to living in a footnote. It seems like a lot of this could be condensed or re-designed, in the case of the figures.

---

> ### Author Rebuttal · Authors · 2023-08-28
>
> We thank the reviewer for the encouraging review.
>
> **(Question A) You state that the data will be made available. How will this be possible if some of the data is behind a paywall, i.e. in the cases where you took paragraphs from Amazon's free preview functionality?**
>
> We appreciate the ethical concerns and would like to clarify that we only use 20 passages for each book (which is often below 1-2% of the book) and release the annotated data for research purposes only. This usage falls under the category of fair use, which we have confirmed via conversations with the HathiTrust organization. We additionally purchased all books ourselves and provided citations for each copy (author, year of publication, translator; see Table 7 for the novels used for translation and Table 9 for the novels used for demonstrations).
>
> **(Question B) Why did you choose the subset of MQM translation span-level errors that you did?**
>
> This is a very valid question. Before conducting the main study, we ran a pilot study to identify which errors are the most prominent and important for our analysis. MQM has many types of errors and was designed to allow the user to select the categories that are relevant to their task (*“MQM is a modular system that allows you to use just a few categories or as many as you need.”* https://themqm.org/faq/). From our pilot study, we observe that the  selected categories cover most if not all issues we saw in the produced translations. Additionally, restricting the number of categories allowed us to better train the translators and reduce their cognitive load.
>
> We appreciate the comments on the writing and paper presentation. We will make the necessary adjustments in the final version of the paper. We also thank the reviewer for pointing out missing citations (from ACL 2023), which we will include in the final version. We would like to clarify that this was not included as the paper comes from ACL 2023 which took place after the submission deadline.

---

### Official Review · Reviewer_KN7M · 2023-08-05

**Typos Grammar Style And Presentation Improvements:** 1. Table-9
**Soundness:** 2

**Excitement:**

2: Mediocre: This paper makes marginal contributions (vs non-contemporaneous work), so I would rather not see it in the conference.

**Missing References:**

1. Kent Chang, et al (2023) Speak, Memory: An Archaeology of Books Known to ChatGPT/GPT-4: https://arxiv.org/abs/2305.00118


**Paper Topic And Main Contributions:**

paper is about comparing performance of LLM-based translation system on literary work.  Authors compare sentence level translation (with no context), sentence level translation with paragraph level context and finally paragraph level translations.  Then they conduct an extensive manual evaluation of results where humans label various kinds of errors.  The conclusion is that paragraph-level translation tends to capture the discourse related elements of the text and all together provides the best translation.

**Questions For The Authors:**

1. Can the success of the LLM-based system (vs. GT) and and its paragraph level translation be attributed to the chance that some of those literary works were part of the LLM pretraining?  Please check the work of Kent Chang et al (2023) on "Archaeology of books known to ChatGPT" which shows the wide range of books that are used to pretrain the LLM.  Based on that ChatGPT might poses a leverage over NMT systems in the literature space.

2. Where does this work leave us at?  anything more than: paragraph-level translation is better than sentence and LLMs still don't a great job on translating literary work?  what kind of new technique and methodology is being offered?

3. Why do you conduct manual tokenization (as mentioned in line 184)?  actually in the caption of table 9 it is mentioned that tools like Spcy have been used.  Please clarify.

4. Why just staying at one paragraph level (considering your many of your current paragraphs are short).  A Natural extension of this analysis would have been to extend the context to 2 or more paragraphs and see how far the improvements take us?  or do analysis on translation quality of longer paragraphs.



**Reasons To Accept:**

1. Focus on literary translation which is a fairly an under-studied area.

2. the annotated data has some unique features and can be useful for the rest of the community



**Reasons To Reject:**

1. The scope and novelty of the scientific work is quite limited.  The paper is suited for a focused workshop on MT or NLP for literary works rather than the main track of the tier-1 EMNLP.

2. Lack of scientific analysis on the why and how questions and unclear future direction of this work.

3. Weak replicability: Due to the black-box nature of the study with the evolving GPT models, it is difficult to replicate these experiments.

**Reproducibility:**

2: Would be hard pressed to reproduce the results. The contribution depends on data that are simply not available outside the author's institution or consortium; not enough details are provided.

**Reviewer Confidence:**

4: Quite sure. I tried to check the important points carefully. It's unlikely, though conceivable, that I missed something that should affect my ratings.

---

> ### Author Rebuttal · Authors · 2023-08-28
>
> We thank the reviewer for the in-depth review and their suggestions.
>
> The reviewer points out that **the scope of the work is limited** as we focus on the evaluation of literary translations. We believe that literary translation is a very specific and important task, which is currently understudied (Toral and Way, 2018 https://link.springer.com/chapter/10.1007/978-3-319-91241-7_12, Thai et al., 2022 https://aclanthology.org/2022.emnlp-main.672/). Literary translation is also more difficult as it requires the translator to not only interpret and convey the author's voice but also make the text enjoyable for the target audience.
>
>
> The reviewer also raises concerns about **the lack of scientific analysis**, which we address in our response to Question 2. In summary, we performed a detailed human evaluation of translation quality, including errors made by each model and each translation approach, resulting in a corpus of annotated errors along with translators’ comments (about 350 hours of effort). We believe that it is important to know how well popular large language models (LLMs) can translate longer inputs, especially since many people are using LLMs for translation without fully understanding their strengths and limitations.
>
>
> The  reviewer also points out that **the paper may suffer from reproducibility issues** as the study employs black-box models. While it is true that it may be hard to reproduce the exact same results due to the black-box nature of the employed models (and model updates), we believe that the general conclusion (i.e., large language models are capable of leveraging longer context resulting in translations better than at the sentence-level) is reproducible, and we saw the same gains with our follow up experiments employing longer texts and newer models (GPT-3.5-turbo and GPT-4). We will update the next version of our paper with more discussion on the translation quality of these two larger-scale LMs.
>
>
> **Q1: Can the success of the LLM-based system (vs. GT) and and its paragraph level translation be attributed to the chance that some of those literary works were part of the LLM pretraining? Please check the work of Kent Chang et al (2023) on "Archaeology of books known to ChatGPT" which shows the wide range of books that are used to pretrain the LLM. Based on that ChatGPT might poses a leverage over NMT systems in the literature space.**
>
> We emphasize that we take the data contamination issue seriously in the paper, as the vast majority of books used in this study are translations published in 2022, which is after the cutoff of GPT-3.5 training data (Jun 2021; https://platform.openai.com/docs/models/gpt-3-5). We further tried to prompt the model to generate completions of the original passages by feeding the model half of the source passage and we observed that the model was unable to generate the exact completion often changing the story completely. We are also currently experimenting with books which have never been translated before (including books published in 2022 and books with no digital copies available) and see similar benefits of the paragraph-level approach.
>
> **Q2: Where does this work leave us at? anything more than: paragraph-level translation is better than sentence and LLMs still don't a great job on translating literary work? what kind of new technique and methodology is being offered?**
>
> Since many users are currently already employing large language models for translation, we believe that it is important to first and foremost understand whether the models have the capability to leverage longer context without significant information or quality loss. Our study provides a detailed analysis of both paragraph and sentence-level translations, including error annotation following the MQM annotation schema, and translators’ comments on each translation. We emphasize that these are novel scientific contributions: the error distribution has previously not been characterized at the paragraph level, and our work shows both where LLMs succeed in using discourse-level context as well as where they need to further improve (we discuss the issues with the current model in section 6 and list detailed errors in tables 15 and 16). We will make this more clear in the next version of our paper. We further release all the evaluation data which can be used for training automatic evaluation metrics for automatic document-level evaluation.
>
> **Q3: Why do you conduct manual tokenization (as mentioned in line 184)? actually in the caption of table 9 it is mentioned that tools like Spcy have been used. Please clarify.**
>
> We tokenized the source sentences manually in order to make sure that there are no mistakes as the current tokenization tools are not equally robust for all the languages employed in the current study. This allowed us to ensure that the sentence-level approach always gets an entire sentence to translate. In order to also provide the reader with a rough estimate of sentences present in translation, we also employed SpaCy/Stanza to tokenize the produced translations. This number, however, is just an estimate and may be inaccurate due to the tokenizer issues (e.g., the tokenizer will work great for English but not necessarily for Japanese). These numbers are meant as rough estimates for the reader and we also provide the number of sentences as per automatic tokenization in the source text for comparison. In contrast, any mistake in the tokenization of the source text for the sentence-level translation could result in providing the model with unfinished sentences (or passages longer than one sentence) affecting the quality of the sentence-level translation.
> We will make sure to clarify it in the writing.
>
> **Q4: Why just staying at one paragraph level (considering your many of your current paragraphs are short). A Natural extension of this analysis would have been to extend the context to 2 or more paragraphs and see how far the improvements take us? or do analysis on translation quality of longer paragraphs.**
>
> We agree with the reviewer that employing even longer text would be more interesting and we are currently working on analyzing this data with more capable models having longer context window. However, GPT-3.5. (davinci-003) employed in this study has a restricted window size of 4k tokens, which after the inclusion of five demonstrations makes it more difficult to fit longer text. This is especially true for languages other than English which usually result in much higher token count (Japanese, for instance, can be three times higher than English) for the same semantic content (see https://arxiv.org/abs/2305.13707). Furthermore, we would like to clarify that many samples do exceed one paragraph as we broadly define a paragraph as a distinct passage within the novel, focusing on a single theme (see footnote 3).
>
> We thank the reviewer for pointing out the missing reference and their suggestions for the paper writing. We will revise the paper accordingly.

---

### Official Review · Reviewer_cWsQ · 2023-08-06

**Soundness:** 4

**Excitement:**

4: Strong: This paper deepens the understanding of some phenomenon or lowers the barriers to an existing research direction.

**Paper Topic And Main Contributions:**

This paper explores how well GPT-3.5 translates literary paragraphs in the few-shot setting (using ICL) in 18 language pairs with varied morphological aspects.


Models:
The authors apply three methods for prompting GPT-3.5 for translation:
SENT: sentence-level translation without context
PARA_SENT: sentence-level translation with source paragraph as context
PARA: paragraph-level translation
They also run Google Translate on the same instances.


How was the evaluation done?
As automatic metrics are not yet reliable, especially for literary paragraphs, they asked native speakers of source and target languages to evaluate the translation performance of GPT-3.5. Following Multidimensional Quality Metrics, annotators were asked to highlight erroneous spans and to classify them to a predefined taxonomy of mistakes. After this step, annotators were asked to identify if the translation contains additions or omissions compared to the source text. Finally, annotators were asked to choose the better translation and justify their choice.


Insights:
- PARA (a method that consists of prompting GPT-3.5 with paragraph-level examples in the prompt) achieves the best performance, even higher than Google Translate
- PARA generates the fewest number of grammatical errors



**Reasons To Accept:**

- Rigorous human evaluation on paragraph-level translation

**Reasons To Reject:**

NA

**Reproducibility:**

3: Could reproduce the results with some difficulty. The settings of parameters are underspecified or subjectively determined; the training/evaluation data are not widely available.

**Reviewer Confidence:**

3: Pretty sure, but there's a chance I missed something. Although I have a good feel for this area in general, I did not carefully check the paper's details, e.g., the math, experimental design, or novelty.

---

> ### Author Rebuttal · Authors · 2023-08-26
>
> We thank the reviewer for the encouraging review.

---

### Meta-Review · Area_Chair_YkMy · 2023-09-02

**Recommendation:** 1
**Confidence:** 4

**Metareview:**

The paper presents a human evaluation of GPT3.5 translating paragraphs/documents of literature across 18 language pairs. The outputs are compared against Google Translate which only works at sentence level. Analysis with MQMs shows interesting findings. Most of the work (350h) was done by professional translators. Annotated data is released.

Pros: This is an interesting analysis that complements previous archival papers and the released annotated data could be useful for future work. Some of the results are good to know. The paper is clearly written and organized.

Cons:  I see a limited scientific contribution with respect to the standards of EMNLP.  1) The type of analysis is not novel and the experimental set up is not scientifically sound:  results are not reproducible, no control of major variables like training data of compared systems, which makes this not an “apple with apple” comparison). The compared systems are black boxes that have been trained independently on data that is not disclosed.
2) The work does not make efforts to improve the status quo, such are improving the design of scientific experiments with third-party LLMs.  As suggested by one reviewer, a better experiment would have been that of comparing document vs. single sentence with one fine-tuned publicly available LLM.

In conclusion, I don’t find   this contribution scientifically very solid and thus not appropriate for a venue like EMNLP. Still, it contains many good-to-know results, that could be valuable for a workshop on translation quality analysis, although the paper focuses on just a single case study.

---

### Decision · Program_Chairs · 2023-10-07

**Decision:**

Reject

**Comment:**

The paper presents a human evaluation of GPT3.5 translating paragraphs/documents of literature across 18 language pairs. The outputs are compared against Google Translate which only works at sentence level. Analysis with MQMs shows interesting findings. Most of the work (350h) was done by professional translators. Annotated data is released.

Pros: This is an interesting analysis that complements previous archival papers and the released annotated data could be useful for future work. Some of the results are good to know. The paper is clearly written and organized.

Cons:  I see a limited scientific contribution with respect to the standards of EMNLP.  1) The type of analysis is not novel and the experimental set up is not scientifically sound:  results are not reproducible, no control of major variables like training data of compared systems, which makes this not an “apple with apple” comparison). The compared systems are black boxes that have been trained independently on data that is not disclosed.
2) The work does not make efforts to improve the status quo, such are improving the design of scientific experiments with third-party LLMs.  As suggested by one reviewer, a better experiment would have been that of comparing document vs. single sentence with one fine-tuned publicly available LLM.

In conclusion, I don’t find   this contribution scientifically very solid and thus not appropriate for a venue like EMNLP. Still, it contains many good-to-know results, that could be valuable for a workshop on translation quality analysis, although the paper focuses on just a single case study.